# Post-retrieval noradrenergic activation impairs subsequent memory depending on cortico-hippocampal reactivation

**Hendrik Heinbockel[1], Gregor Leicht[2], Anthony D Wagner[3], Lars Schwabe[1]***

[1]Department of Cognitive Psychology, University of Hamburg, Hamburg, Germany; [2]Department of Psychiatry and Psychotherapy, University Medical Center Hamburg Eppendorf, Hamburg, Germany; [3]Department of Psychology and Wu Tsai Neurosciences Institute, Stanford, United States

## eLife Assessment

This work presents **important** findings of a modulatory effect of yohimbine, an alpha2-adrenergic antagonist that raises noradrenaline levels, on the reconsolidation of emotionally neutral word-picture pairs, depending on the hippocampal and cortical reactivation during retrieval. The evidence supporting the main conclusions is **convincing**, with an elegant design combining fMRI and psycho-pharmacology. The work will be of broad interest to researchers working on memory.

***For correspondence:**
lars.schwabe@uni-hamburg.de

**Competing interest:** The authors declare that no competing interests exist.

**Abstract** When retrieved, seemingly stable memories can become sensitive to significant events, such as acute stress. The mechanisms underlying these memory dynamics remain poorly understood. Here, we show that noradrenergic stimulation after memory retrieval impairs subsequent remembering, depending on hippocampal and cortical signals emerging during retrieval. In a three-day study, we measured brain activity using fMRI during initial encoding, 24 hr-delayed memory cueing followed by pharmacological elevations of glucocorticoid or noradrenergic activity, and final recall. While post-retrieval glucocorticoids did not affect subsequent memory, the impairing effect of noradrenergic arousal on final recall depended on hippocampal reactivation and category-level reinstatement in the ventral temporal cortex during memory cueing. These effects did not require a reactivation of the original memory trace and did not interact with offline reinstatement during rest. Our findings demonstrate that, depending on the retrieval-related neural reactivation of memories, noradrenergic arousal after retrieval can alter the future accessibility of consolidated memories.

## Introduction

Memories are not stable entities but can undergo changes long after initial consolidation (*Nadel et al., 2012*; *Dudai and Eisenberg, 2004*). The updating of existing memories in light of new information or experiences is a key feature of adaptive memory. A potential mechanism underlying such updating is memory reconsolidation. According to reconsolidation theory, memories become labile again upon their reactivation, requiring another period of stabilization (i.e. reconsolidation; *Nader and Einarsson, 2010*; *Lee et al., 2017*; *Schwabe et al., 2014*). During the reconsolidation window, memories are assumed to be modifiable (*Galarza Vallejo et al., 2019*; *Kroes et al., 2014*). Alternative views posit that post-reactivation changes in memory are due to the emergence of new traces during retrieval, potentially interfering with the retrieval of the original memory (*Nadel et al., 2000*; *Sederberg et al., 2011*; *Polyn et al., 2009*). The dynamics of memory after retrieval, whether through reconsolidation of the original trace or interference with retrieval-related traces, have fundamental implications for

educational settings, eyewitness testimony, or mental disorders (*Schwabe et al., 2014*; *Clem and Schiller, 2016*; *Schacter and Loftus, 2013*). In clinical contexts, post-retrieval changes of memory might offer a unique opportunity to retrospectively modify or render less accessible unwanted memories, such as those associated with posttraumatic stress disorder (PTSD) or anxiety disorders (*Walsh et al., 2018*; *Xue et al., 2012*; *Björkstrand et al., 2016*). Given these potential far-reaching implications, understanding the mechanisms underlying post-retrieval dynamics of memory is essential.

Stress has a major impact on memory (*de Quervain et al., 1998*; *Roozendaal et al., 2009*; *Schwabe et al., 2022*). While most studies have focused on stress effects on memory formation or retrieval, accumulating evidence suggests that stress may also alter the dynamics of memory after retrieval. The majority of studies suggest a disruptive influence of post-retrieval stress on subsequent remembering (*Dongaonkar et al., 2013*; *Schwabe and Wolf, 2010b*; *Maroun and Akirav, 2008*; *Larrosa et al., 2017*; *Hupbach and Dorskind, 2014*, but see *Bos et al., 2014b*; *Coccoz et al., 2011* for an opposite effect). Although post-retrieval stress-induced changes in putative memory reconsolidation or accessibility are highly relevant in legal or clinical contexts, the mechanisms involved in these effects remain poorly understood. Recently, we showed a detrimental impact of post-retrieval stress on subsequent memory that was contingent upon reinstatement dynamics in the Hippocampus, VTC, and PCC during memory reactivation (*Heinbockel et al., 2024*). While this study provided initial insights into the potential brain mechanisms involved in the effects of post-retrieval stress on subsequent memory, the underlying neuroendocrine mechanisms remained elusive.

It is well known that stress triggers complex neurotransmitter and hormonal cascades (*Joëls and Baram, 2009*). Among these, noradrenaline and glucocorticoids appear to be of particular relevance for stress-induced changes in memory (*de Quervain et al., 1998*; *Roozendaal et al., 2006a*; *Strange and Dolan, 2004*). Pharmacological studies in humans and rodents demonstrate a significant impact of noradrenaline and glucocorticoids on the posited reconsolidation or mnemonic interference processes after retrieval. However, their exact roles in post-retrieval memory dynamics are unclear. Some studies, using emotional recognition memory or fear conditioning in healthy humans, suggest enhancing effects of post-retrieval glucocorticoids on subsequent memory (*Antypa et al., 2019*; *Meir Drexler et al., 2015*). However, rodent studies on neutral recognition memory (*Maroun and Akirav, 2008*), fear conditioning (*Vafaei et al., 2023*), as well as evidence from humans on episodic recognition memory (*Antypa et al., 2021*) report impairing effects of glucocorticoid receptor activation on post-retrieval memory dynamics. For noradrenaline, post-retrieval blockade of noradrenergic activity impairs putative reconsolidation or future memory accessibility in human fear conditioning (*Kindt et al., 2009*), as well as drug (alcohol) memory (*Schramm et al., 2016*) and spatial memory in rodents (*Przybyslawski et al., 1999*). However, this effect is not consistently observed in human studies on fear conditioning (*Bos et al., 2014a*), speaking anxiety (*Elsey et al., 2020*), inhibitory avoidance (*Muravieva and Alberini, 2010*), traumatic mental imagination (PTSD patients) (*Wood et al., 2015*), and might depend on the arousal state of the individual (*Maroun and Akirav, 2008*) or the exact timing of drug administration as suggested by studies in humans (*Thomas et al., 2017*) and rodents (*Otis et al., 2014*). Thus, while there is evidence that glucocorticoid and noradrenergic activation after retrieval can affect subsequent memory, the direction of these effects remains elusive. Moreover, the brain mechanisms underlying the potential effects of post-retrieval glucocorticoids or noradrenergic arousal on subsequent remembering are largely unknown, especially in humans.

Extant studies suggest that brain regions implicated in initial memory formation, such as the hippocampus, may also play a role in the modification of memories after their reactivation (*Nader et al., 2000*; *Przybyslawski and Sara, 1997*; *Schwabe et al., 2012b*). Research in transgenic mice indicates that effective post-reactivation interventions require the reactivation of specific neuronal subsets within the engram, underscoring the significant contribution of the original memory trace to changes during the proposed reconsolidation window (*Khalaf et al., 2018*). While human neuroimaging studies cannot assess the reactivation of individual neurons within an engram, multivariate pattern analysis (MVPA) enables the assessment of neural pattern reinstatement at the stimulus category or event level (*Staresina et al., 2012*; *Wing et al., 2015*; *Thakral et al., 2015*; *Polyn et al., 2005*; *Kuhl et al., 2011*). Notably, memory reactivation occurs not only during goal-directed retrieval (online) but also offline during post-retrieval rest periods. Online reactivation reflects the immediate impact of memory retrieval on neural networks and may involve modifications of the existing memory trace and/or the encoding of a new memory trace in response to retrieval demands (*Tanaka et al.,*

*2014*; *Johnson and Rugg, 2007*). Offline reactivation offers a pivotal window for the consolidation and stabilization of these memory alterations (*Tambini et al., 2010*; *Oudiette and Paller, 2013*; *Staresina et al., 2013*). The transition from online to offline reactivation involves complex neural cascades, influencing the persistence and strength of the reactivated memory trace (*Yagi et al., 2023*). Fundamental knowledge gaps remain about the role of online and offline neural reactivation in post-retrieval dynamics of human memory in general, and its modulation by stress mediators in particular.

This pre-registered study aimed to elucidate the brain mechanisms underlying the impact of post-retrieval glucocorticoids and noradrenaline on subsequent remembering in humans, with a specific focus on whether the effects of post-retrieval stress are contingent on online or offline neural reinstatement. To this end, healthy participants underwent a three-day experiment. On Day 1, participants encoded a series of word-picture pairs and subsequently completed an immediate cued recall test. On Day 2 (24 hr later), half of the learned words were presented again during a Memory Cueing task, prompting participants to consciously retrieve the associated pictures and thereby reactivate their underlying neural representations. Notably, according to both reconsolidation and interference accounts of post-retrieval changes in memory (*Nader and Einarsson, 2010*; *Sederberg et al., 2011*), only cued items that were reinstated should be susceptible to post-retrieval manipulations. We distinguished between responses with short and long reaction times indicative of high and low confidence responses because previous research showed that reaction times are inversely correlated with hippocampal memory involvement (*Starns, 2021*; *Robinson et al., 1997*; *Ratcliff and Murdock, 1976*) and memory strength (*Weidemann and Kahana, 2016*; *Gimbel and Brewer, 2011*), and that high confidence memories associated with short reaction times may be particularly sensitive to stress effects (*Gagnon et al., 2019*). The remaining words served as non-reactivated controls. Importantly, shortly before the Memory Cueing task, participants received orally either a Placebo (N=20), 20 mg Hydrocortisone (N=21), or 20 mg of the α2-adrenoceptor antagonist yohimbine (N=21) leading to increased noradrenergic stimulation. This timing of drug administration was chosen to result in significant elevations of glucocorticoid or noradrenergic activity after completion of the Memory Cueing task, during the proposed post-retrieval consolidation or reconsolidation window. The action of the drugs was assessed by arousal and salivary cortisol measured before and after drug intake. On Day 3 (another 24 hr later), participants underwent a final cued recall memory test, enabling assessment of the impact of post-retrieval noradrenergic and glucocorticoid activation on subsequent memory performance.

Critically, brain activity was recorded using fMRI throughout all stages of the memory paradigm, on all three days. On Day 2, we also included resting-state scans before and after the Memory Cueing task to assess offline memory reactivation. Given that associative memories rely on the hippocampus and cortical representation areas (*Ranganath et al., 2004*; *Kim, 2010*), such as the ventral temporal cortex (VTC), which represents stimulus categories (scenes, objects) encountered during encoding (*Bracci et al., 2017*; *Grill-Spector and Weiner, 2014*), and the posterior cingulate cortex (PCC), which is assumed to represent memory traces formed during retrieval (*Thakral et al., 2015*; *Bird et al., 2015*), we focused our analysis on these key regions. Building on our recent findings in humans (*Heinbockel et al., 2024*) as well as current insights from rodents (*Staresina et al., 2012*), we hypothesized that the effects of post-retrieval noradrenergic and glucocorticoid activation would critically depend on the reinstatement of the neural event representation during retrieval. To investigate memory reinstatement, we employed multivariate pattern analysis (MVPA) and representational similarity analysis (RSA) across experimental days.

Here, we show that pharmacological elevations of noradrenergic but not glucocorticoid activity after retrieval impair subsequent remembering. These memory impairments were specific to items that were cued and correctly recalled behaviourally and associated with strong hippocampal and cortical neural reactivation before the action of yohimbine; that is, the mere cueing and behavioural expression of memories were not sufficient to render memories sensitive to modification. As such, our neural data revealed that the disruptive effects of yohimbine on subsequent memory were contingent on the strength of hippocampal reactivation and category-level pattern reinstatement in the VTC during memory retrieval. Critically, the impact of noradrenaline specifically depended on the preceding online reactivation, as the level of offline reactivation prior to drug activation did not impact subsequent memory.

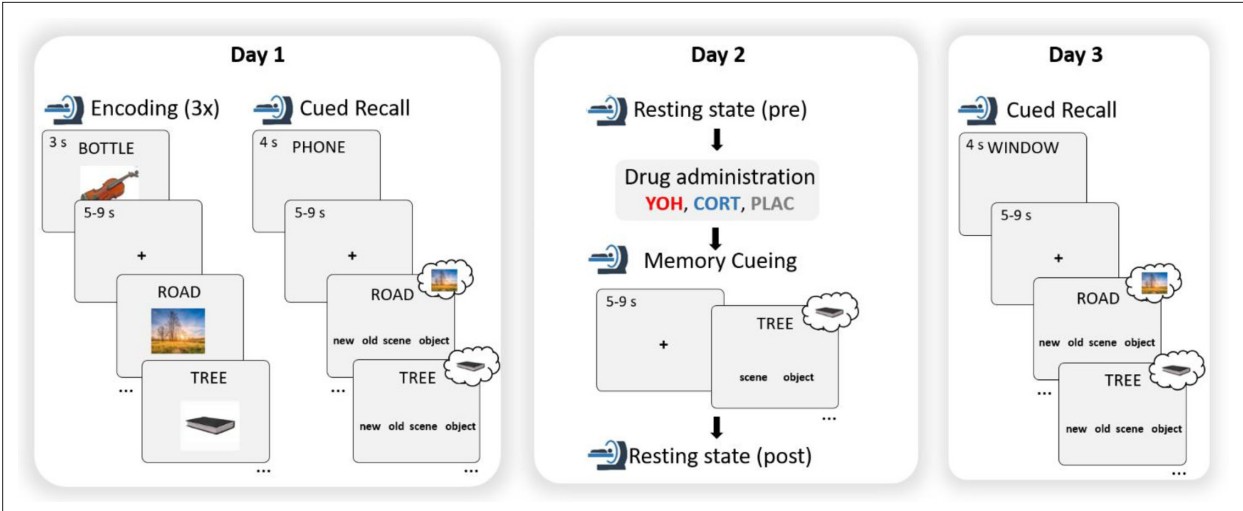

**Figure 1.** Experimental task. The impact of post-retrieval yohimbine and hydrocortisone on subsequent memory was tested in a 3 d paradigm, recording fMRI data on all days. On Day 1, participants encoded word-picture pairs across three runs and then underwent an immediate cued recall test. On Day 2, 24 hr later, participants started with a 10 min resting-state fMRI scan, followed by the oral administration of 20 mg yohimbine (YOH), 20 mg hydrocortisone (CORT), or a placebo (PLAC). Thereafter, in the Memory Cueing task, half of the word-picture pairs were cued by presenting the corresponding word; Day 2 ended with another 10 min resting-state scan. On Day 3, again 24 hr later, participants completed a final cued recall test including word cues for all 144 pairs from Day 1 encoding, half of which had been cued and half of which had not been cued on Day 2, along with 152 new foils.

## Results

### Day 1: Successful memory encoding

After completing an associative encoding task comprising 164 word-picture pairs (*Figure 1*), participants engaged in an immediate cued recall task in which 144 previously presented 'old' word cues (plus eight catch trials) were presented intermixed with 152 'new' foils. On each trial, participants could respond with one of four options: 'old/scene,' 'old/object,' 'old,' or 'new' (4AFC decision; *Figure 1*). Participants successfully distinguished between old words and new words, with a 74.4% hit rate (response 'old,' 'old/scene,' 'old/object' to an old word) and a 16.8% false alarm rate (response 'old,' 'old/scene,' 'old/object' to a new word). Participants recognized the word and correctly identified the associated image category in 47.3% of trials (associative category hit rate) with an associative error rate of 13.1%. Signal detection theory-based analysis revealed an average associative d' of 1.13 (SE = 0.09).

Because the critical stress system manipulations were implemented only on Day 2 (hydrocortisone, yohimbine, placebo groups), we confirmed that immediate cued recall performance on Day 1 did not differ between pairs later cued and uncued on Day 2 (F(1,58) = 1.25, p=0.267, $\eta^2$<0.01), nor between groups (all main or interaction effects; all Ps>0.481; see *Appendix 1—table 1*). To test for potential group differences in reaction times for correctly remembered associations on Day 1, we fit a linear model including the factors *Group* and *Cueing*. Critically, we did not observe a significant *Group × Cueing* interaction, suggesting no RT difference between groups for later cued and not cued items (F(2,58) = 1.41, p=0.258, $\eta^2$=0.01; *Appendix 1—table 1*). Moreover, groups did not differ in mood, arousal, or cortisol levels before encoding on Day 1 (all Ps>0.564; see *Appendix 1—table 2*). Whole-brain fMRI analyses on immediate cued recall data (associative category hits >associative misses), considering the within-subject factor *Cued* and the between-subjects factor *Group,* revealed no significant main or interaction effects (all Ps>0.564; see *Appendix 1—table 2*). These outcomes suggest comparable neural underpinnings of immediate (Day 1) memory retrieval for pairs that, on Day 2, were subsequently cued and correctly remembered and pairs subsequently uncued, as well as across experimental groups on Day 1.

## Day 2: Neural signatures of successful memory reactivation
### Successful memory cueing

On Day 2, participants returned to the MRI scanner for a Memory Cueing task (cued recall; 2AFC; *Figure 1*) in which half of the word-picture associations encoded on Day 1 were cued. Before the Memory Cueing task, there were no significant differences between groups in subjective mood, autonomic arousal, or salivary cortisol (all Ps>0.096, *Appendix 1—table 2*). During this task, participants were presented with 76 old cue words (36 previously paired with scenes, 36 previously paired with objects, and four catch trials). Participants were instructed to recall the picture associated with the word cue in as much detail as possible and to indicate whether the picture depicted an object or a scene. Due to the absence of new foils in this task, memory outcomes were restricted to associative hits (i.e. correct trials) and associative misses (i.e. incorrect trials). Overall, participants performed well, accurately identifying the correct picture category in 67.5% of trials (SE = 2.6%; chance = 50%), and the three groups did not differ in performance (F(2,58) = 1.53, p=0.224, $\eta^2$=0.05; see *Appendix 1—table 1*). To test for potential group differences in reaction times for correctly remembered associations on Day 2, we fit a linear model including the factors *Group* and *Reaction time (slow/fast)* following the subject-specific median split. The model did not reveal any main effect or interaction including the factor *Group* (all Ps>0.535; *Appendix 1—table 1*), indicating that there was no RT difference between groups, nor between low and high RT trials in the groups.

### Neural reactivation in hippocampus and cortical areas during memory cueing

Drawing upon recent discoveries in rodent studies (*Khalaf et al., 2018*), we hypothesized that the impact of post-retrieval noradrenergic and glucocorticoid activation would hinge significantly on the reactivation of neural event representations during and after retrieval. To initially elucidate the neural underpinnings of successful memory retrieval (i.e. retrieval success), we examined univariate brain activity on associative hits vs. associative misses in the Memory Cueing task. A whole-brain fMRI analysis revealed significant activation in bilateral hippocampi (Left: [-26,–32, –10], t=7.93, P(FWE)<0.001; Right: [32, -40, -12], t=7.89, P(FWE)<0.001), ventral temporal cortex (VTC; Left: [-30,–40, –12], t=7.75, P(FWE)<0.001; Right: [52, -50, -14], t=7.26, P(FWE)<0.001), and PCC ([4, -42, 38], t=8.10, P(FWE)<0.001), along with other regions central for episodic memory retrieval (e.g. medial prefrontal cortex; see *Appendix 1—table 3*). Importantly, there were no group differences in univariate brain activity related to successful retrieval during the Memory Cueing task (all *Retrieval success × Group* interaction Ps >0.420).

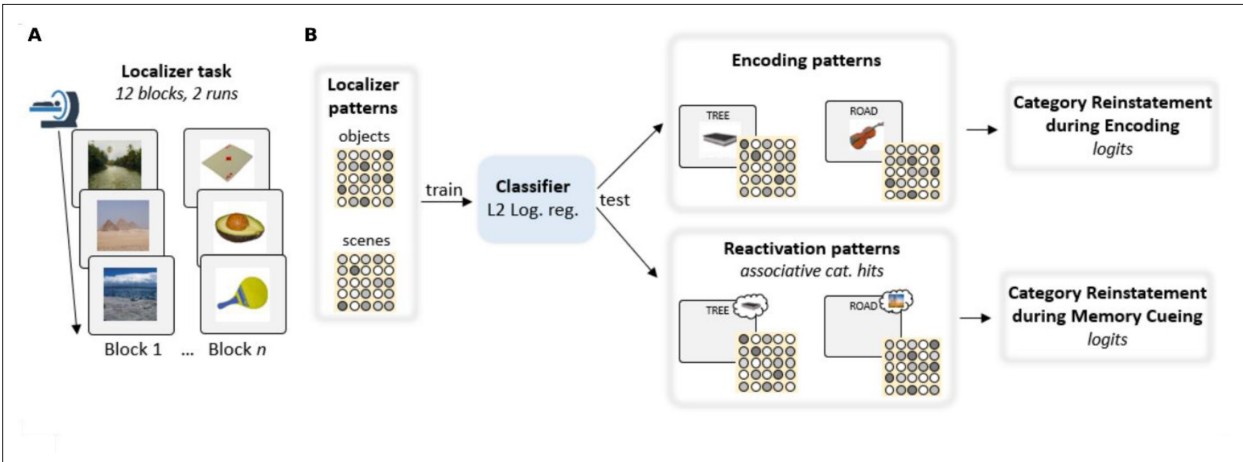

**Figure 2.** Trial-wise pattern reinstatement during Encoding and the Day 2 Memory Cueing task. (**A**) To derive an index of visual category reinstatement in the ventral temporal cortex (VTC), an independent localizer task was conducted at the end of Day 3. During this task, pictures of scenes and objects were presented block-wise to participants. (**B**) The resulting neural patterns of both categories were then used to train an L2-regularized logistic regression. This function served to classify trial-wise patterns during the Day 1 Encoding task as well as the Day 2 Memory Cueing task, while also providing the strength of category-level online reinstatement (quantified as logits). Associative cat. hits = associative category hits.

A linear mixed-effects model (LMM) using participants' reaction times as a proxy for memory confidence/memory strength revealed that higher hippocampal as well as PCC activity was associated with faster 2AFC reaction times (Left hippocampus: $\beta$=–0.51 ± 0.18, t=–2.88, $P_{Corr}$ = 0.018, $R^2_{conditional}$ = 0.08; Right hippocampus: $\beta$=–0.47 ± 0.18, t=–2.60, $P_{Corr}$ = 0.033, $R^2_{conditional}$ = 0.11; PCC: $\beta$=–0.75 ± 0.20, t=–3.67, $P_{Corr}$ <0.001, $R^2_{conditional}$ = 0.09), while no such relation was observed in the VTC ($P_{Corr}$ = 0.282). Importantly, LMMs did not reveal main or interaction effects including the factor *Group* (all Ps>0.131). Thus, while these four regions were generally more active during successful vs. unsuccessful memory cueing, activity in the hippocampus and PCC also tracked memory confidence/memory strength (also shown in *Gordon et al., 2014*).

## Category-level pattern reinstatement in hippocampus and cortical areas during memory cueing

In an independent localizer task, we assessed the discriminability of category-related beta patterns in the VTC, hippocampus, and PCC while participants viewed scenes, faces, and objects (*Figure 2A*). Employing a leave-one-run-out cross-validated L2-regularized logistic regression analysis, we classified scenes versus objects and evaluated classifier performance based on accuracy. Classifier accuracy is derived from the sum of correct predictions the trained classifier made in the test-set, relative to the total amount of predictions. For the VTC, the average classifier accuracy was high (M ± SD: 90.0% ± 0.1%); t(60) = 25.99, $P_{Corr}$ <0.001, d=3.83 indicative of reliable category-level processing in the VTC. Importantly, there were no significant group differences in classification accuracy (F(1,59) = 2.56, $P_{Corr}$ = 0.115, $\eta^2$=0.04). Further probing VTC category processing, we next tested the localizer-trained classifier on the Day 1 Encoding task (*Figure 2B*), in which objects and scenes were presented. Average accuracy was again high (M ± SD: 77.9% ± 0.9%, t(60) = 29.88, $P_{Corr}$ <0.001, d=3.80), further supporting category-level processing in the VTC, again without significant group differences in classification accuracy (F(2,58) = 0.44, $P_{Corr}$ = 0.643, $\eta^2$=0.01).

Next, we quantified the reinstatement of visual category-level representations during successful memory cueing on Day 2 in the VTC. Neural reinstatement reflects the extent to which a neural activity pattern (i.e. for objects) that was present during encoding is reactivated during retrieval (e.g. memory cueing). Using the localizer-trained logistic classifier, testing on all trials of the Memory Cueing task (in which only words but not associated images were presented) confirmed that associative hits were accompanied by stronger visual category pattern reinstatement in VTC, compared to associative misses (main effect *Retrieval Success*: F(1,58) = 12.45, $P_{Corr}$ <0.001, $\eta^2$=0.13). Importantly, there were no significant differences between groups in VTC reinstatement during the Memory Cueing task (all main and interaction effects, Ps >0.504). Subsequently, we tested whether the strength of single-trial category-level reinstatement (logits) in VTC was predicted by Day 2 memory performance. The logits here reflect the log-transformed trial-wise probability of a pattern either representing a scene or an object. A generalized linear mixed model revealed a main effect of *Retrieval success* (F(1,58) = 12.61, $P_{Corr}$ = 0.003, $\eta^2$=0.13), but no effect of *Group* and no *Group ×Retrieval success* interaction (both Ps = 1), showing that successful memory cueing on Day 2 was associated with greater trial-wise category-level reinstatement in the VTC, without differences between groups. Finally, we tested the VTC-trained classifier selectively on associative hit trials, corresponding to remembered scenes and objects, during the Memory Cueing task. Overall, the classifier distinguished remembered scenes from remembered objects, performing significantly above chance-level (50%; M ± SE = 54.4% ± 1.0%; t(60) = 4.44, $P_{Corr}$ <0.001, d=1.14), without a difference between scenes and objects (p=0.092). By contrast, when tested on associative miss trials, the classifier failed to differentiate forgotten scenes from forgotten objects (M ± SE = 50.1% ± 1.7%; $P_{Corr}$ = 1). Again, classifier accuracy on remembered trials in VTC did not differ between groups (F(2,58) = 0.86, $P_{Corr}$ = 1, $\eta^2$=0.03).

We also examined scene vs. object classification accuracy in the left and right hippocampus, using data from the independent localizer. The average accuracy scores did not significantly differ from chance (50%; Left: M ± SD: 53.3% ± 1.8%, t(60) = 1.72, $P_{Corr}$ = 0.501, d=0.22; Right: M ± SD: 52.9% ± 1.5%, t(60) = 1.50, $P_{Corr}$ = 0.520, d=0.18), indicating poor category-coding in the hippocampus (*Liang et al., 2013*). We also trained the classifier on the localizer runs (scenes vs. objects) and tested it on the Day 1 Encoding task data, in which objects and scenes were presented. The average accuracy scores were above chance-level (50%; Left: M ± SD: 53.8% ± 0.9%, t(60) = 3.29, $P_{Corr}$ = 0.006, d=0.42; Right: M ± SD: 53.4% ± 0.9%, t(60) = 3.71, $P_{Corr}$ <0.001, d=0.93) indicating category-coding in

the hippocampus during visual encoding, without significant group differences in classification accuracy (Left: F(1,59) = 0.02, p=.874, $\eta^2$<.01; Right: F(1,59) = 0.03, $P_{Corr}$ = .784, $\eta^2$<0.01). However, in contrast to VTC, classifiers trained on localizer activation patterns in the left and right hippocampus were neither able to distinguish remembered scenes and remembered objects (Left: M ± SE = 50.71% ± 1.0%; t(60) = 0.69, $P_{Corr}$ = 1, d=0.09; Right: M ± SE = 51.82% ± 0.9%; t(60) = 2.10, $P_{Corr}$ = 0.156, d=0.23), nor forgotten scenes and forgotten objects (Left: M ± SE = 47.95% ± 1.6%; t(60) = −1.31, $P_{Corr}$ = 1, d=0.17; Right: M ± SE = 49.61% ± 1.3%; t(60) = −0.27, $P_{Corr}$ = 1, d=0.09) when tested on Day 2 Memory Cueing task data.

Finally, we examined scene vs. object classification accuracy in the PCC using localizer task data. The average accuracy scores significantly exceeded chance level (50%; M ± SD: 62.4% ± 2.24%, t(60) = 5.39, $P_{Corr}$ <0.001, d=0.69), indicating category-coding in PCC, without group differences (F(1,59) = 0.81, $P_{Corr}$ = 0.370, $\eta^2$=0.01). We also trained the classifier on the localizer runs (scenes vs. objects) and tested it on the Day 1 Encoding task data. The average accuracy scores were above chance (50%; M ± SD: 54.6% ± 1.0%, t(60) = 4.43, $P_{Corr}$ <0.001, d=0.57), indicating category-coding in the PCC during visual encoding, with no significant group differences in classification accuracy (F(1,59) = 0.45, $P_{Corr}$ = 1, $\eta^2$<0.01). The classifier trained on localizer activation patterns in the PCC was neither able to distinguish remembered scenes and remembered objects during the Day 2 Memory Cueing task (M ± SE = 52.3% ± 0.98%; $P_{Corr}$ = 0.092), nor forgotten scenes and forgotten objects (M ± SE = 49.5% ± 1.70%; t(60) = −0.27, $P_{Corr}$ = 1, d=0.03).

Contrasting within-localizer classifier accuracies revealed a main-effect of *Region* (F(2,174) = 101.74, $P_{Corr}$ <0.001, $\eta^2$=0.054). Post-hoc tests revealed significantly higher accuracy for the VTC compared to PCC (t(60) = −12.00, $P_{Corr}$ <0.001, d=1.54) and hippocampus (t(60) = −17.40, p<0.001, d=2.24), and for the PCC compared to the hippocampus (t(60) = −3.90, $P_{Corr}$ <0.001, d=0.50). Moreover, while we found evidence for category-level reinstatement during Day 1 Encoding in the VTC, PCC, and hippocampus, a main-effect of *Region* (F(2,174) = 192.32, p<.001, $\eta^2$=0.69) revealed significantly higher accuracy for the VTC compared to PCC (t(60) = −16.90, $P_{Corr}$ <0.001, d=2.18) and hippocampus (t(60) = −19.01, $P_{Corr}$ <0.001, d=2.45). Classifier accuracy of PCC and hippocampus did not differ during the Encoding task (t(60) = 0.94, $P_{Corr}$ = 1, d=0.12). Finally, significant category-level reinstatement of remembered trials during the Day 2 Memory Cueing task was observed in cortical areas (VTC, PCC), but not in the hippocampus. Comparing corresponding accuracy estimates revealed a main-effect of *Region* (F(2,174) = 3.45, p=0.034, $\eta^2$=0.04). Post-hoc tests showed no difference between VTC and PCC (t(60) = −1.69, $P_{Corr}$ = 0.283, d=0.22) nor PCC and hippocampus (t(60) = 1.24, $P_{Corr}$ = 0.660, d=0.16), whereas VTC accuracy was significantly higher than hippocampal accuracy (t(60) = −2.61, $P_{Corr}$ = 0.034, d=0.34).

## No evidence for event-level online reinstatement

Beyond category-level reinstatement, we assessed event-level memory trace reinstatement from initial encoding (Day 1) to memory cueing (Day 2), via RSA, correlating neural patterns in each region (hippocampus, VTC, and PCC) across days. To test for evidence that associative hits during memory cueing entailed the reinstatement of representations established at encoding, we compared the average event-level Day 1 (encoding) to Day 2 (memory cueing) similarity of the associative hits against 0. In PCC and hippocampus, we did not obtain evidence for event-level memory trace reinstatement (t-test against 0; both $P_{Corr}$ >0.296). By contrast, for the VTC, average similarity was significantly negative, suggesting that from Day 1 (encoding) to Day 2 (memory cueing), neural patterns became more dissimilar (t(60) = −7.87, $P_{Corr}$ <0.001, d=1.01). As the VTC is implicated in category-level processing, we next compared trial-wise event- vs category-level similarities. Results revealed that memory trace reinstatement during successful memory cueing on Day 2 (i.e. associative hits) was characterized by significantly higher category-level representations compared to event-level representations in all three regions (hippocampus: t(60) = 5.51, $P_{Corr}$ <0.001, d=0.71; VTC: t(60) = −11.83, $P_{Corr}$ <0.001, d=1.51; PCC: t(60) = 8.25, $P_{Corr}$ <0.001, d=1.06). This outcome is consistent with the above MVPA outcomes demonstrating that associative hits on Day 2 are accompanied by category-level reinstatement (as quantified by the localizer-trained classifier). Given this finding, all subsequent analyses focused on category-level, rather than event-level, patterns.

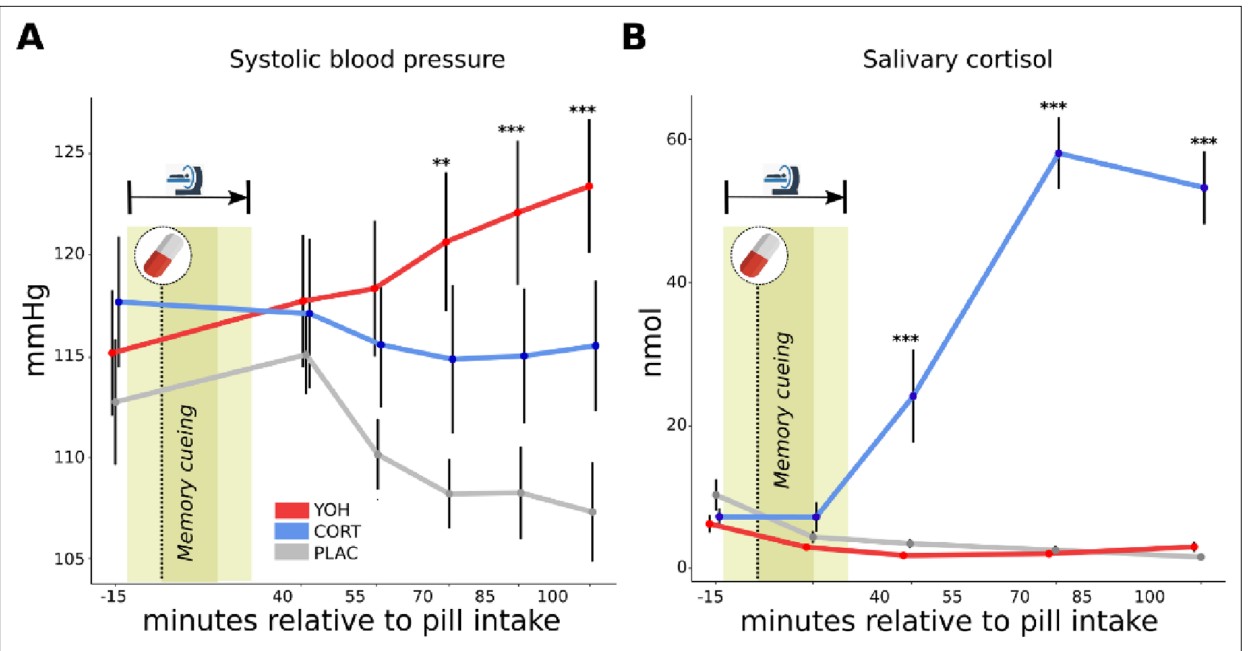

**Figure 3.** ffective noradrenergic and glucocorticoid action after Day 2 memory cueing. Systolic blood pressure (**A**) and salivary cortisol (**B**) did not differ between groups before or immediately after the Memory Cueing task. However, 70 min after pill intake, systolic blood pressure was significantly higher in the yohimbine (YOH; n = 21) group relative to the placebo (PLAC; n = 20) group. Conversely, salivary cortisol was significantly higher in the hydrocortisone (CORT; n = 21) group relative to PLAC starting 40 min after pill intake. Light yellow shades indicate the pre- and post-memory cueing resting-state fMRI scan periods. Data represent means (±SE). ***p<0.001, **p<0.01.

### Day 2: Noradrenergic activity and glucocorticoid concentrations

Shortly before the Memory Cueing task, participants were administered either 20 mg YOH (n=21), 20 mg CORT (n=21), or a PLAC (n=20). Given the known pharmacodynamics of YOH and CORT, we expected the drugs to be effective after the Memory Cueing task and subsequent resting-state interval (*Krenz et al., 2021*; *Kluen et al., 2017*), exerting their influence during the putative post-retrieval (re)consolidation window. To confirm successful noradrenergic and glucocorticoid activation, and to verify that their effects occurred only after (but not during) the Memory Cueing task, we assessed autonomic arousal (blood pressure, heart rate, and skin conductance), salivary cortisol, and subjective mood throughout Day 2.

Analysis of autonomic measures revealed a significant Time × *Group* interaction in systolic blood pressure (F(8.71, 256.99)=5.87, p<0.001, $\eta^2$=.03; *Figure 3A*), but not in diastolic blood pressure or heart rate (both Ps >0.120; *Appendix 1—table 6*). Post-hoc t-tests showed significantly higher systolic blood pressure in the YOH group compared to the PLAC group 70 min (t(29.77)=−3.31, $P_{Corr}$ = 0.014, d=1.02), 85 min (t(34.15)=−3.33, $P_{Corr}$ = 0.012, d=1.03), and 100 min after pill intake (t(36.94)=−3.98, $P_{Corr}$ <0.001, d=1.23). The CORT group did not significantly differ from the PLAC group in systolic blood pressure (all Ps >0.229). Importantly, systolic blood pressure in the YOH and CORT groups did not differ from the PLAC group immediately before or after the MRI session, suggesting that the drug was not yet effective during the Memory Cueing task and the post-reactivation resting-state scan (both Ps >0.485).

We also recorded skin conductance, a continuous indicator of autonomic arousal, during the MRI scans (i.e. during the Memory Cueing task and the resting-state scans), when the drug should not have been active yet. Skin conductance response analysis during the Memory Cueing task and pre- and post-reactivation resting-state scans showed no Time × *Group* interaction (F(3.30, 97.44)=0.33, p=0.819, $\eta^2$<0.01) and no main effect of *Group* (F(2,59) = 2.60, p=0.083, $\eta^2$=0.07), suggesting that groups did not reliably differ in autonomic arousal during the MRI scans.

In contrast to systolic blood pressure, salivary cortisol increased, as expected, in the CORT group but not in the YOH or PLAC groups (Time × *Group* interaction: F(5.33, 157.17)=43.80, p<0.001, $\eta^2$=0.472). Post-hoc t-tests indicated a significant cortisol increase in the CORT group compared to

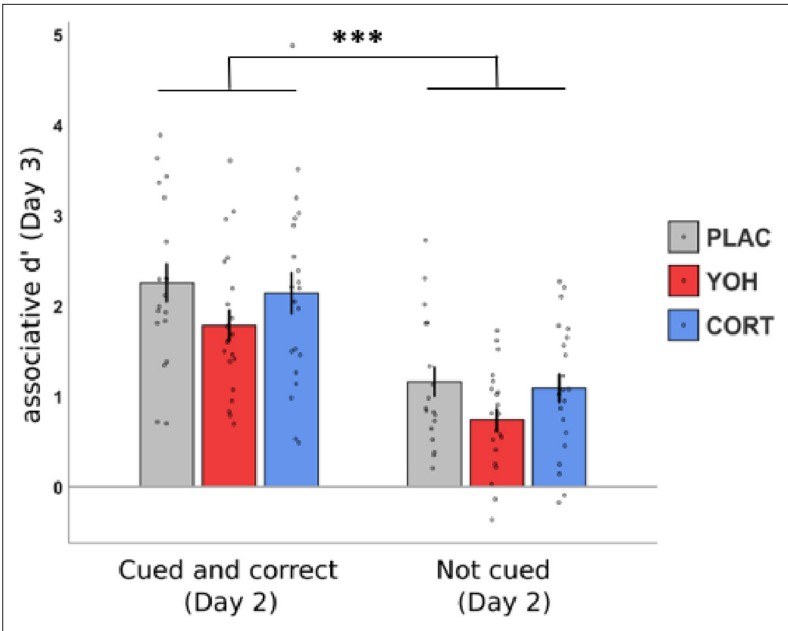

**Figure 4.** Subsequent memory performance on Day 3, split for cued and correct (Day 2), and uncued trials. Average memory performance (associative d') was significantly increased for cued and correct (Day 2) trials compared to uncued trials. This effect was, however, unaffected by the pharmacological manipulation. Data represent means (± SE). ***p<0.001.

the PLAC group at 40 min (t(27.91)=2.30, $P_{Corr}$ = 0.020, d=0.99), 70 min (t(20.64)=–11.23, $P_{Corr}$ <0.001, d=3.42), and 100 min after pill intake (t(20.19)=–10.36, p<0.001, d=3.15; *Figure 3B*), whereas salivary cortisol of PLAC and YOH groups revealed no significant difference at any timepoint (all $P_{Corr}$ >0.350). Importantly, salivary cortisol concentrations did not differ between groups immediately before or during the MRI session, suggesting that CORT was not yet effective during the Memory Cueing task or post-reactivation resting-state scan (both $P_{Corr}$ >0.162). Finally, subjective mood analyses across Day 2 revealed no significant Time × *Group* interaction on any scale (all interaction Ps >0.460; *Appendix 1—table 7*). Our data demonstrate that, despite the distinct pharmacodynamics of CORT and YOH, both substances are active within the time window that is critical for potential reconsolidation effects (*Nader and Einarsson, 2010*; *Lee et al., 2017*; *Nader et al., 2000*).

## Day 3: Memory cueing increases subsequent memory performance

On Day 3, 24 hrafter memory cueing and drug administration, participants returned to the MRI scanner for a final cued recall task. Groups did not differ in subjective mood, autonomic arousal, or salivary cortisol before this final memory test (all Ps>0.158, see *Appendix 1—table 2*). The Day 3 cued recall task was identical to that on Day 1, except that it contained novel lures. Participants successfully distinguished between old words and new words, with an 81.1% hit rate (response 'old,' 'old/scene,' 'old/object' to an old word) and a 21.75% false alarm rate (response 'old,' 'old/scene,' 'old/object' to a new word). Participants recognized the word and correctly identified the associated image category in 50.1% of trials (associative category hit rate) with an associative error rate of 11.6%. Day 3 associative d' was 1.14 (SE = 0.15). Importantly, across groups, memory was significantly enhanced for associations that were cued and successfully retrieved on Day 2 (M=2.05; SE = 0.21) compared to uncued associations (M ± SE = 1.14 ± 0.15; F(1,58) = 143.51, p<0.001, $\eta^2$=0.29; *Figure 4*; see *Appendix 1—table 1*), in line with the established testing effect (*Roediger and Karpicke, 2006*; *Karpicke and Roediger, 2008*), and confirming the efficacy of the selective, association-specific cueing manipulation.

According to both memory reconsolidation and mnemonic interference accounts, drugs should selectively affect subsequent memory for associations cued and reactivated before the effective action of the drugs on Day 2 but not for uncued items. When collapsing across all cued associations (i.e. not considering whether the memory was indeed reactivated), a mixed-design ANOVA on associative d' scores revealed neither a significant Cued × *Group* interaction nor a main effect of *Group*

(all Fs <2.08, all Ps>0.134), suggesting that the mere presentation of the word cue on Day 2 was insufficient to induce post-retrieval stress hormone effects that change future memory performance. To test for potential group differences in reaction times for correctly remembered associations on Day 3 we fit a linear model including the factors *Group* and *Cueing*. This model did not reveal any main effect or interaction including the factor *Group* (all Ps>0.267), indicating that there was no average RT difference between groups. As expected we observed a main effect of the factor *Cueing*, indicating a significant difference in reaction times across groups between trials that were successfully cued and those not cued on Day 2 (F(2,58) = 153.07, p<0.001, $\eta^2$=0.22; *Appendix 1—table 1*). Furthermore, univariate analyses showed no *Cued × Group* interactions in whole-brain or ROI activity.

### Day 3: Effects of post-retrieval noradrenergic stimulation on subsequent memory depend on prior online hippocampal and cortical reactivation

We hypothesized that the post-retrieval effects of noradrenergic arousal and cortisol on subsequent memory depend on robust neural memory reactivation shortly before the action of the drugs on Day 2. We, therefore, tested whether the strength of neural reactivation during successful memory cueing (Day 2) predicted the impact of post-retrieval noradrenergic and glucocorticoid activation on subsequent memory (Day 3). Overall, univariate activity on cued and correct trials (Day 2 associative hits) in the hippocampus, PCC, and VTC did not reveal any interaction with *Group* on subsequent memory (Day 3 associative d'), suggesting that the average activation across trials and voxels within a single brain area may not suffice to predict post-retrieval effects of noradrenaline or cortisol (all interaction $P_{Corr}$ >0.711).

Reaction times in the Day 2 Memory cueing task revealed a trial-specific gradient in reactivation strength. Thus, we turned to single-trial analyses, differentiating Day 3 trials by short and long reaction times during memory cueing on Day 2 (median split), indicative of high vs. low memory confidence (*Starns, 2021*; *Robinson et al., 1997*; *Ratcliff and Murdock, 1976*) and hippocampal reactivation (*Heinbockel et al., 2024*; *Gagnon et al., 2019*). A GLMM was employed to predict associative category hits on Day 3 by *Group* and *Day 2 Reaction time (short, long)*. A significant interaction (*Group × Reaction time (Day 2)* interaction: $\beta$=0.79 ± 0.30, z=2.61, p=0.008, $R^2_{conditional}$ = 0.27; *Figure 5A*)

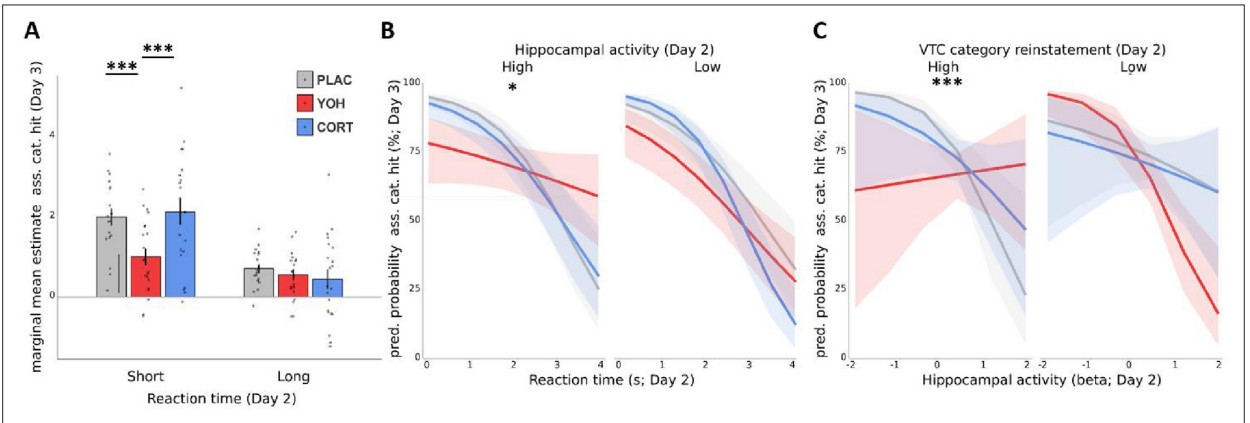

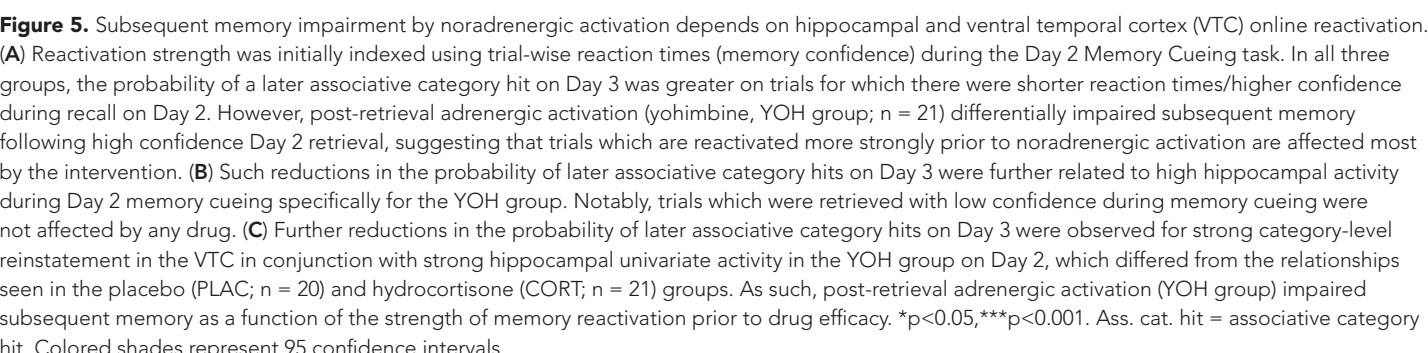

**Figure 5.** Subsequent memory impairment by noradrenergic activation depends on hippocampal and ventral temporal cortex (VTC) online reactivation. (**A**) Reactivation strength was initially indexed using trial-wise reaction times (memory confidence) during the Day 2 Memory Cueing task. In all three groups, the probability of a later associative category hit on Day 3 was greater on trials for which there were shorter reaction times/higher confidence during recall on Day 2. However, post-retrieval adrenergic activation (yohimbine, YOH group; n = 21) differentially impaired subsequent memory following high confidence Day 2 retrieval, suggesting that trials which are reactivated more strongly prior to noradrenergic activation are affected most by the intervention. (**B**) Such reductions in the probability of later associative category hits on Day 3 were further related to high hippocampal activity during Day 2 memory cueing specifically for the YOH group. Notably, trials which were retrieved with low confidence during memory cueing were not affected by any drug. (**C**) Further reductions in the probability of later associative category hits on Day 3 were observed for strong category-level reinstatement in the VTC in conjunction with strong hippocampal univariate activity in the YOH group on Day 2, which differed from the relationships seen in the placebo (PLAC; n = 20) and hydrocortisone (CORT; n = 21) groups. As such, post-retrieval adrenergic activation (YOH group) impaired subsequent memory as a function of the strength of memory reactivation prior to drug efficacy. *p<0.05,***p<0.001. Ass. cat. hit = associative category hit. Colored shades represent .95 confidence intervals.

revealed that the relationship between Day 2 reactivation and the probability of an associative hit on Day 3 varied across groups. Post-hoc marginal means tests revealed a differential decrease in the probability of associative hits on Day 3 in light of short Day 2 reaction times when comparing YOH vs. CORT ($\beta$=2.55 ± 0.94, z-ratio=2.55, $P_{Corr}$ = 0.031) and YOH vs. PLAC ($\beta$=0.34 ± 0.14, z-ratio=–2.55, $P_{Corr}$ = 0.032). By contrast, comparing CORT vs. PLAC revealed no such difference ($\beta$=0.88 ± 0.37, z-ratio=–0.29, $P_{Corr}$ = 1), suggesting that noradrenergic arousal specifically interacts with strongly reactivated representations after retrieval.

As hippocampal and PCC activity scaled with Reaction times from the Day 2 Memory cueing task, we next differentiated trials according to the strength of their neural reactivation. To relate Day 2 reactivation strength to subsequent memory (Day 3), we fit GLMMs, predicting Day 3 associative category hits by *ROI activity* (Day 2), *Reaction time* (Day 2), and *Group*. Strikingly, shorter reaction times and stronger hippocampal activity on Day 2 predicted an increased probability of an associative category hit on Day 3 memory in the PLAC group, whereas these measures of stronger reactivation on Day 2 predicted a lower probability of an associative category hit on Day 3 in the YOH group (*Group × Hippocampal activity (Day 2) × Reaction time (Day 2)* interaction: $\beta$=0.90 ± 0.36, z=2.45, $P_{Corr}$ = 0.038, $R^2_{conditional}$ = 0.27) but not in the CORT group ($\beta$=0.89 ± 0.39, z=2.28, $P_{Corr}$ = 0.068). Post-hoc comparisons confirmed significant differences in strongly reinstated trials between YOH and PLAC groups ($\beta$=–1.12 ± 0.35, z-ratio=–3.13, $P_{Corr}$ = 0.005) and between YOH and CORT groups ($\beta$=0.88 ± 0.34, z-ratio=2.58, $P_{Corr}$ = 0.029), but not between PLAC and CORT groups ($\beta$=–0.23 ± 0.36, z-ratio=–0.63, $P_{Corr}$ = 1; *Figure 5B*). Parallel models with univariate PCC and right hippocampal activity did not yield a significant interaction with *Group* (all Ps>0.081), suggesting that cued memories specifically accompanied by left hippocampal reactivation during Day 2 was associated with increased vulnerability to the influence of post-retrieval YOH, disrupting post-retrieval processing and subsequent memory on Day 3.

We further hypothesized that the post-retrieval effects of noradrenergic arousal and cortisol on subsequent memory would depend on the reinstatement of the original memory trace (as assayed by the similarity of neural patterns during Encoding and Memory Cueing). We, therefore, tested whether the strength of memory trace reinstatement in the hippocampus, VTC, and PCC during successful memory cueing (Day 2) predicted the impact of post-retrieval noradrenergic and glucocorticoid activation on subsequent memory (Day 3). In contrast to our prediction, none of these regions showed a significant effect that included the factor *Group* (all $P_{Corr}$ >0.257). These results suggest that the previously observed post-retrieval noradrenergic subsequent memory impairment may be associated with retrieval-related univariate activity but not the reinstatement of encoding-related neural patterns.

Building on our observation that category-level pattern reinstatement during Day 2 memory cueing (assessed by MVPA) in the VTC was linked to successful memory retrieval, we next classified cued and correct (Day 2) trials as strongly or weakly reactivated based on a median-split on the strength of VTC category-level pattern reinstatement (assayed by logits), allowing us to include the uncued trials in further analyses. Testing whether *Reactivation strength (uncued, low VTC reinstatement, high VTC reinstatement)* interacted with *Group* and *Hippocampal activity (Day 2)* to predict Day 3 (24-hr-delayed) memory performance yielded a significant interaction ($\beta$=–0.21 ± 0.07, z=–3.08, p=0.002, $R^2_{conditional}$ = 0.18; *Figure 5B*). Post-hoc slope tests confirmed that noradrenergic activation significantly affected Day 3 memory for the trials associated with stronger trial-wise VTC category-level pattern reinstatement and hippocampal univariate activity on Day 2, resulting in an impairment of subsequent retrieval on Day 3 (YOH vs. PLAC: $\beta$=0.14 ± 0.05, z-ratio=2.57, $P_{Corr}$ = 0.030; YOH vs. CORT: $\beta$=0.13 ± 0.05, z-ratio=1.31, $P_{Corr}$ = 0.708; *Figure 5C*). By contrast, neither drug affected Day 3 memory for the trials associated with weaker trial-wise VTC category-level pattern reinstatement and hippocampal univariate activity on Day 2 (all Ps>0.210). Notably, when directly comparing the slopes of weak and strong category-level VTC reinstatement in interaction with hippocampal activity, only the YOH group showed a significant decrease related to Day 3 performance (YOH: $\beta$=0.12 ± 0.05, z-ratio=2.72, $P_{Corr}$ = 0.018; CORT: β = - 0.02±0.05, z-ratio=–0.42, $P_{Corr}$ = 1; PLAC: β = - 0.09±0.05, z-ratio=–1.68, $P_{Corr}$ = 0.274). As individual factors, such as metabolism or body weight, can influence the drug's action, we ran an additional analysis in which we included individual (baseline-to-peak) differences in salivary cortisol and (systolic) blood pressure, respectively. This analysis did not show any group by baseline-to-peak difference interaction suggesting that the observed memory effects were mainly driven by the pharmacological intervention group per se and less by individual variation

in responses to the drug (see Appendix Inter-individual differences in pharmacological responses did not significantly mediate subsequent memory).

## Explorative analyses

Beyond hippocampal and VTC activity during memory cueing (Day 2), we exploratively reanalysed the GLMMs predicting Day 3 memory performance including the PCC, which was relevant during memory cueing in the current study and in our previous work (*Heinbockel et al., 2024*). Predicting Day 3 memory performance by the factors *Group* and *Single trial beta activity* during memory cueing in the PCC did not reveal a significant interaction ($P_{Corr}$ = 1); adding the factor *Reaction time* to the model also did not result in a significant interaction ($P_{Corr}$ = 1). We also included the medial prefrontal cortex (MPFC) to predict Day 3 memory performance, as the MPFC has been shown to be sensitive to noradrenergic modulation in previous work (*Radley et al., 2008*). Predicting Day 3 memory performance by the factors *Group* and *Single trial beta activity* during memory cueing in the MPFC did not reveal a significant interaction ($P_{Corr}$ = 1); adding the factor *Reaction time* to the model also did not result in a significant interaction ($P_{Corr}$ = 1), which indicates that the MPFC was not modulated by either pharmacological intervention. Finally, we investigated memory cueing from all remaining ROIs that were significantly activated during the Day 2 memory cueing task (Day 2 whole-brain analysis; correct-incorrect; *Appendix 1—table 3*). We again fit GLMMs predicting Day 3 memory performance by the factors *Group* and *Single trial beta activity* during memory cueing. Again, we did not observe any significant interaction effect on any of the ROIs (all interaction $P_{Corr}$ >0.060) and these results did not change when adding the factor *Reaction time* to the respective models (all $P_{Corr}$ >0.075).

## Connectivity analyses

We conducted general psycho-physiological interaction analysis (gPPI) analyses on the Day 2 memory cueing task (remembered – forgotten), which revealed that successful cueing was accompanied by significant functional connectivity between the left hippocampus, VTC, PCC, and MPFC (see *Appendix 1—table 4*). However, using these connectivity estimates to predict Day 3 subsequent memory performance (dprime) via regression did not reveal any significant Group × *Connectivity* interactions, indicating that the pharmacological manipulation (i.e. noradrenergic stimulation) did not modulate subsequent memory based on functional connectivity during memory cueing (all $P_{Corr}$ >0.228). The same pattern of results was observed when including single-trial beta estimates from multiple ROIs during memory cueing to predict Day 3 memory (all interaction effects $P_{Corr}$ >0.288).

## Offline reinstatement analyses

Aside from examining neural activity related to retrieval during the Memory Cueing task, we also investigated offline reactivation, which is manifested in neural reinstatement observed during the resting-state scans conducted both pre and post-memory cueing (Appendix Tracking offline reactivation). Neural representations from the Memory Cueing task were reinstated significantly offline (i.e. post >pre resting state) in the hippocampus, PCC, and VTC. Moreover, the initial patterns from encoding were reinstated offline in the VTC (Appendix Offline encoding and cueing reinstatement in hippocampus and cortical areas). However, in contrast to the above-reported online reactivation × drug effects, none of these factors interacted with the *Group* when considering Day 3 subsequent memory performance (Appendix Day 3 offline pattern reactivation does not modulate effects of post-retrieval noradrenergic activation on subsequent memory performance).

## Discussion

Upon their retrieval, memories can become sensitive to modification (*Nadel et al., 2012*; *Dudai and Eisenberg, 2004*). Such post-retrieval changes in memory may be fundamental for adaptation to volatile environments and have critical implications for eyewitness testimony, clinical or educational contexts (*Schwabe et al., 2014*; *Clem and Schiller, 2016*; *Schacter and Loftus, 2013*; *Walsh et al., 2018*; *Xue et al., 2012*; *Björkstrand et al., 2016*), Yet, the brain mechanisms involved in the dynamics of memory after retrieval are largely unknown, especially in humans. Here, we aimed to shed light on the neural mechanisms underlying the impact of post-retrieval elevations in the major stress mediators noradrenaline and cortisol on subsequent remembering. Our results revealed that post-retrieval

noradrenergic activation led to an impairment in subsequent memory, depending on memory strength/confidence, hippocampal activation, and VTC pattern reinstatement during memory reactivation. By contrast, post-retrieval glucocorticoid activation did not influence subsequent memory in any way.

Previous research showed that administering the beta-blocker propranolol after memory reactivation reduces subsequent memory, potentially interfering with the putative reconsolidation process (*Kindt et al., 2009*; *Schramm et al., 2016*; *Przybyslawski et al., 1999*; *Schwabe et al., 2012a*). While this impairing influence has not been consistently replicated (*Bos et al., 2014b*; *Elsey et al., 2020*; *Wood et al., 2015*; *Muravieva and Alberini, 2010*), these results suggest that post-retrieval noradrenaline may facilitate subsequent remembering. In contrast to this idea, our results demonstrate that increased noradrenergic stimulation after memory retrieval impairs subsequent memory. However, a key distinction between our study and prior research using propranolol lies in the emotional nature of the memory task. Previous studies predominantly focused on emotionally arousing information or fear memories (*Lee et al., 2006*; *Debiec and Ledoux, 2004*; *Phelps et al., 2004*), assuming that post-retrieval propranolol may weaken reconsolidation by attenuating the emotional salience of memories, making them more comparable to neutral ones (*Schwabe et al., 2012b*). In our study, we employed emotionally neutral scene images, offering a novel context to explore noradrenergic effects on memory (re)consolidation or mnemonic interference. Furthermore, our findings suggest a potential inverted u-shaped relationship between post-retrieval noradrenergic arousal and subsequent memory, where both noradrenergic blockade by propranolol and strong noradrenergic stimulation induced by yohimbine result in a subsequent memory impairment. This idea is in line with previous reports of inverted U-shaped relationships between noradrenergic arousal and memory processes (*Arnsten, 2011*; *Hernaus et al., 2017*; *Birnbaum et al., 1999*; *Li and Mei, 1994*). Most importantly, our results suggest that the yohimbine-induced memory impairment critically depended on hippocampal reactivation during memory cueing. The hippocampus, crucial for episodic memory formation and retrieval, is highly sensitive to noradrenergic modulation, which can impact hippocampal long-term potentiation and depression (*Strange et al., 2014*; *Katsuki et al., 1997*). Excessive noradrenergic activity in the hippocampus may further have disrupted neurotransmission (*Kim and Kim, 2023*; *Diamond et al., 2007*). This disruption may have manifested as deficits in consolidating new retrieval-related memory traces or reconsolidating existing memories. Furthermore, the subsequent memory impairment in the YOH group was additionally dependent on robust activation of the VTC during memory cueing. These effects could relate to an impeding of the (re)consolidation of visual memory contents, given the VTC's role in processing complex visual stimuli and encoding categorical information, such as scenes (*Bracci et al., 2017*; *Grill-Spector and Weiner, 2014*).

In addition to noradrenergic activation, acute stress is accompanied by a significant increase in cortisol levels, which has been associated with impairments in putative memory reconsolidation after retrieval (*Maroun and Akirav, 2008*; *Vafaei et al., 2023*; *Antypa et al., 2021*; *Wang et al., 2008*). Our results revealed that post-retrieval glucocorticoid activation did not influence subsequent memory, as the placebo and cortisol groups performed similarly in the subsequent memory task. Acute stress triggers a series of neurochemical changes, and it has been shown that noradrenergic and glucocorticoid activation are strongly intertwined. Accordingly, previous studies have highlighted that the effects of glucocorticoids on memory processes are particularly pronounced when accompanied by high noradrenergic arousal, commonly observed during stressful situations (*Schwabe et al., 2022*; *Roozendaal et al., 2006a*; *de Quervain et al., 2007*). Notably, in the current study, the administration of hydrocortisone was not associated with an increase in arousal or negative mood. As such, our findings may imply that cortisol alone is not sufficient to influence post-retrieval updating and necessitates concurrent noradrenergic arousal for its memory-modulating effects to fully manifest (*Maroun and Akirav, 2008*; *Roozendaal et al., 2006b*).

There is evidence suggesting that memory updating depends not only on neural processes during retrieval (i.e. online processing) but also on offline neural reinstatement or replay during post-retrieval rest (*Staresina et al., 2013*; *Schlichting and Preston, 2014*). However, whether offline neural reinstatement after retrieval is involved in post-retrieval changes of subsequent memory remains unclear. Here, we tested for the first time whether post-retrieval manipulations of memory are dependent on neural offline reinstatement after memory cueing. While we generally observed significant offline reactivation events in the post-cueing interval compared to pre-cueing (see Appendix Offline encoding and cueing reinstatement in hippocampus and cortical areas), our findings revealed that neither drug

significantly affected subsequent memory via interacting with offline reinstatement dynamics (see Appendix Day 3 offline pattern reactivation does not modulate effects of post-retrieval noradrenergic activation on subsequent memory performance). To explain this absence of an effect, it is important to note the differences between the estimated neural online compared to offline parameters. While it might seem that offline reinstatement reflects a mere repetition of the neural signal reactivated during retrieval, these two parameters are not directly comparable.

We investigated offline reactivation in the brain during rest periods before and after a Memory Cueing task by examining neural patterns with RSA. We compared neural activity from the Memory Cueing task with resting-state fMRI scans taken before and after the task, focusing on the hippocampus, VTC, and PCC. To identify reactivation events, we calculated the mean correlation plus 1.5 standard deviations from the pre-cueing phase and applied this threshold to assess pre- and post-cueing correlation matrices. We repeated this process using the post-cueing threshold. Finally, we quantified the number of offline reactivation events by counting the correlations that exceeded these thresholds. The reported offline reinstatement events are hence based on differences in correlations that exceed a threshold, and do not reflect the direct strength of the underlying neural correlate (such as e.g. trial-wise hippocampal activity). Given that the observed impairments in subsequent memory in the YOH group were directly dependent on the trial-specific strength of online neural reactivation (i.e. hippocampal activity and reaction times) one would need to derive a comparable assay from the offline intervals. Finally, on that matter is important to note that the reaction time (confidence) during memory cueing was the most powerful predictor of post-retrieval effects; a predictor that can not be derived from resting state intervals.

In line with the central tenets of reconsolidation theory (*Nader and Einarsson, 2010*; *Lee et al., 2017*; *Schwabe et al., 2014*), the disruptive effects of YOH were contingent on memory reactivation. There were no differential effects of noradrenergic activation on cued but incorrectly recalled events relative to uncued events, suggesting that memories, if not correctly recalled, remained resistant to modification. Moreover, the extent of neural reactivation on Day 2 correlated with subsequent memory performance, further underlining the crucial role of neural memory reactivation for post-retrieval modifications of memory. Notably, the triggering of putative reconsolidation is posited to be initiated by prediction errors (PEs; *Diaz-Mataix et al., 2013*; *Sevenster et al., 2013*; *Fernández et al., 2016*). In the present study, PEs may have resulted from the incomplete reminder structure during cued recall (*Kroes et al., 2014*; *Sinclair and Barense, 2019*). That said, our findings are more in line with the disruption of the consolidation of retrieval-related memory presentations rather than reconsolidation theory, as we did not observe interactions of any drug with the reinstatement of the original memory trace. Thus, the observed effects of post-retrieval noradrenaline on subsequent remembering were potentially owing to alterations in new memory traces formed during retrieval, as suggested by multiple trace theory (or interference) accounts of post-retrieval changes in memory. This interpretation is speculative and limited by the fact that we also did not observe any drug interactions with pattern reconfigurations across days.

Finally, it is important to note that we administered drugs before memory cueing on Day 2, in order to achieve, in light of the known pharmacodynamics of hydrocortisone and yohimbine (*Krenz et al., 2021*; *Schwabe et al., 2010a*), effective drug actions shortly after memory reactivation, during the proposed (re)consolidation window. However, as we administered drugs before memory cueing, these could have potentially affected the memory reactivation itself, rather than post-retrieval processes. Our physiological data indicated that the drugs were effective only after the Memory Cueing task. Moreover, groups did not significantly differ in performance or associated neural activity in the Memory Cueing task. These data support the assumption that the drugs did not interfere with memory cueing or reactivation processes, but rather most likely affected post-retrieval (re)consolidation processes.

Previous research demonstrated that acute stress after retrieval, during the proposed reconsolidation window, can impair subsequent memory (*Dongaonkar et al., 2013*; *Schwabe and Wolf, 2010b*; *Maroun and Akirav, 2008*; *Larrosa et al., 2017*; *Hupbach and Dorskind, 2014*). Here, we show that post-retrieval increases of noradrenergic arousal, but not of cortisol, which reduce subsequent remembering. Critically, the observed memory impairment depended on the strength of online neural reinstatement occurring during retrieval, but not offline reinstatement after retrieval, especially in the hippocampus and neocortical representation areas. Our findings provide novel insights into the mechanisms involved in post-retrieval dynamics of memory in general and in those involved in the impact

of stress mediators after retrieval in particular. Beyond their theoretical relevance, these findings may have relevant implications for attempts to employ post-retrieval manipulations to modify unwanted memories in anxiety disorders or PTSD (*Parsons and Ressler, 2013*; *Wessa and Flor, 2007*). Specifically, the present findings suggest that such interventions may be particularly promising if combined with cognitive or brain stimulation techniques ensuring a sufficient memory reactivation.

## Materials and methods

This study was preregistered before the start of data collection at the *German Clinical Trials Register* (DRKS; https://drks.de/search/en/trial/DRKS00029365).

### Participants

Sixty-eight healthy, right-handed adults (28 women, 40 men) without a life-time history of any neurological or psychiatric disease were recruited for this experiment. Further exclusion criteria comprised smoking, drug abuse, prescribed medication use, pregnancy or lactation, a history of kidney- or liver-related diseases, body-mass index below 19 or above 26 kg/m², diagnosed cardiovascular problems as well as any contraindications for MRI measurements. Women were excluded if they used hormonal contraceptives and were not tested during their menses as these factors may interact with the pharmacological intervention. Participants were instructed to refrain from caffeinated beverages, exercise, and eating or drinking (with the exception of water) for 2 hr prior to the experiment. Seven participants were excluded from analyses due to acute claustrophobia (n=1) or technical failure (n=3), no Day 3 memory performance (n=1), or because they did not return on Day 2 or 3 (n=2), thus leaving a final sample of n=61 participants (25 women, 36 men, age = 19–34 y, mean = 25 y, SD = 4 y). We employed a PLAC-controlled, double-blind, between-subjects design in which participants were randomly assigned to one of three groups: PLAC, YOH, or CORT. All participants provided written informed consent before the start of the experiment and received a monetary compensation for their participation. An a priori power calculation with G*Power (*Faul et al., 2007*) indicated that a sample size of N=66 is required to detect a medium-sized Group × *Reactivation* interaction effect with a power of.95. The study was approved by the ethics committee of the Medical Chamber of Hamburg (PV5960). Groups did not differ significantly from each other with respect to depressive mood, chronic stress, state or trait anxiety (see Appendix Control variables and *Appendix 1—table 7*).

### Experimental procedure

The study took place on three consecutive days, with all tasks conducted in the MRI scanner during morning hours (8:30 am - 12:30 pm) to control for the diurnal rhythm of cortisol. On each day, we obtained measures of blood pressure, heartrate, salivary cortisol, and mood to control for potential baseline differences between groups as well as to assess the effective pharmacological manipulation on Day 2.

### Experimental day 1: Associative encoding task

Participants underwent a brief (~5 min) training session before the encoding task to familiarize themselves with the procedure. This training replicated the 3 d paradigm structure, involving an encoding session and a cued recall test with word-picture associations that were not part of the actual experiment. In the actual encoding task, participants were instructed to memorize 164 unique word-picture pairs presented in three runs. Each pair appeared three times (once in each run), including German nouns (see Appendix Stimulus material) and pictures of colored scenes (*Xiao et al., 2010*) or objects (*Brodeur et al., 2014*). During each trial, a word and picture were presented for 3 s (words on top of the screen, pictures in the centre), and participants rated their fit on a 4-point Likert scale using an MRI-compatible button box. A black fixation cross appeared between trials for 5–9 s (jitter: 0–4 s, mean-jitter: 2 s). Each run took about 25 min, with a 2 min break after each run, resulting in a duration of about 90 min for the three runs. Importantly, out of the 164 word-picture pairs presented during encoding, 20 pairs were designated as catch trials for the subsequent cued recall tasks (see Appendix Catch trials). As such, all memory analyses were based on 144 of the encoded word-picture pairs.

### Experimental day 1: Immediate cued recall

After the encoding task, participants were provided a 15 min break before receiving instructions for the immediate cued recall task. Back in the MRI scanner, 152 words (including eight catch trials)

from the prior study phase ('old') and 152 new words were presented. Each test word appeared for 4 s, prompting participants to make one of four memory decisions: 'new,' 'old,' 'old/scene,' or 'old/object.' The latter two responses were used upon recognizing the word as old and indicating the associated images category. Responses were made using an MRI-compatible button box. The positions of 'old/scene' and 'old/object' were randomized (50%) between the ring and little fingers on each trial. Between trials, there was an ITI of 5–9 s (jitter: 0–4 s, mean jitter: 2 s), during which a black fixation cross was presented. The task lasted 60 min in total, divided into two 30 min sessions with a 2 min break in between.

## Experimental day 2: Drug administration and memory cueing

On Day 2, participants returned to the MRI scanner and initially underwent 10 min of eyes-open resting state scanning. Next, participants received orally one of the pharmacological agents (YOH, CORT) or a PLAC, depending on the experimental group. YOH is an α2-adrenoceptor antagonist that leads to increased adrenergic stimulation, while CORT is the synthetic variant of the stress hormone cortisol. The timing and dosage of the drugs were chosen in accordance with previous studies (*Kausche et al., 2021*; *Zerbes et al., 2022*). We note that yohimbine and hydrocortisone follow distinct pharmacodynamics (*Tam et al., 2001*; *Werumeus Buning et al., 2017*), yet selected the administration timing to ensure that both substances are active within the relevant post-retrieval time window. They were taken orally under the supervision of the experimenter immediately before the Memory Cueing task, in order to ensure the action of the drug shortly after the reactivation, i.e., during the reconsolidation window. The pills were indistinguishable, and the experimenter remained unaware of the participants' group assignments, ensuring double-blind testing. Following pill intake, participants completed a Memory Cueing task, which lasted about 20 min. The task included half of the previously studied old words (72 trials, 36 word-scene associations, and 36 word-object associations) and four catch-trials. The words from Day 1 were re-presented for 4 s, with an ITI of 5–9 s (jitter: 0–4 s, mean-jitter: 2 s). On each trial, participants were asked to remember the specific picture that had been associated with this word (i.e. the retrieval cue) during the Day 1 encoding session. Participants were requested to indicate the category of the picture belonging to the presented word. The position of the response options (objects vs. scene; category level 2AFC) was randomly switched between the ring and little fingers on each trial. Because the task was 2AFC for categories, hits, and misses could reflect correct/incorrect retrieval of the associated category but also could reflect recognition of the word as old and a correct/incorrect guess about the associated category remembered or a failure to recognize the word along with a correct/incorrect category guess. It is for this very reason that the neural measures of memory reactivation are incisive, as they provide a means of differentiating 2AFC associative hits that were based on strong associative memory reactivation from those based on moderate reactivation from those based on little to no reactivation. Examining the gradient between stronger and weaker reactivation is also pivotal for understanding the impact of post-retrieval interventions on memory processes, as a strong reactivation during Day 2 may make the memory more susceptible to the effects of pharmacological agents. This task aimed to reactivate half of the word-picture pairs, allowing examination of 'testing effects' and potentially opening a reconsolidation window. The remaining half of the pairs were not reactivated and served as baseline/control memories. After the Memory Cueing task, another 10- min, eyes-open resting state scan was performed. Participants were then taken out of the scanner and led into a separate room were they were seated for 1 hr (provided with magazines to read) while they completed mood questionnaires and we took physiological measurements (e.g. blood pressure) to validate the action of the drugs.

To assess the efficacy of the pharmacological manipulation and the temporal dynamics of the drug action, we measured systolic and diastolic blood pressure, heart rate, salivary cortisol (Sarstedt, Germany), and subjective mood before drug administration (baseline), after the post-reactivation resting-state scan (40 min) and then in four further intervals of 15 min (55, 70, 85, 100 min after drug intake). In order to verify that neither agent would take effect during the critical Memory Cueing task, we additionally obtained a saliva sample directly after the Memory Cueing task (25 min) and recorded the heartrate as well as skin conductance rate continuously throughout the three MRI sessions. Saliva samples were stored at −20 °C until the end of the study. From saliva, we analyzed the free fraction of cortisol by means of a luminescence assay (IBL, Germany). Inter- and intra-assay coefficients of variance were below 10%.

## Experimental day 3: Cued recall and functional localizer

Twenty-four hours after the reactivation session, participants returned to the MRI unit for the final cued recall task, which was identical to the immediate cued recall task on Day 1. Again, participants were presented 152 of the encoded words and 152 new words in random order and were asked to indicate for each word, whether it was 'new,' 'old,' or 'old' and presented with a scene ('old/scene') or 'old' and presented with an object ('old/object'). Following the final cued recall task, participants completed two runs of a visual category localizer task inside the MRI scanner, which served to later identify subject-specific patterns of category-level visual representations (especially in VTC). This task involved judgments about images from three categories: faces (CFD database *Ma et al., 2015*), objects (BOSS database *Brodeur et al., 2014*), and scenes (SUN database *Xiao et al., 2010*). Ten pictures of each category were presented in twelve blocks (4 blocks per picture category) and repeated in two runs. Categories were randomly switched between blocks. During each block, a picture was presented for 0.5 s, with an ITI of 1 s. During the image presentation, participants had to judge whether in case of scenes, it was 'indoor' or 'outdoor,' in the case of objects it was 'artificial' or 'living,' and in case of faces whether it was 'female' or 'male.' Upon completion of the first run, a 1 min break was provided. The second run included the exact same blocks as the first, block-categories were however randomly mixed again.

## Behavioral memory data analysis

In our examination of word-picture associative memory during the cued recall tasks on Day 1 and Day 3 (4AFC), associative category hits were recorded when participants correctly matched old word cues with the corresponding picture category (e.g. responding 'old/scene' for a scene associate), indicating recognition of the presented word as old and retrieval of the associated picture category at the category level. Associative category errors occurred when an old word was recognized, but the wrong category was chosen (e.g. responding to 'old/object' for a scene associate). We use the term 'associative misses' to encompass all old trials that did not result in associative category hits (i.e. an old word was presented and the participant responded 'new,' 'old,' or 'old' with the wrong category). The average rates of associative category hit, misses, and errors were calculated based on correct/incorrect responses relative to the total number of cued and correct (Day 2 Memory Cueing task) and non-cued trials.

During the 2AFC Memory Cueing task on Day 2, participants could only select 'scene' or 'object' as responses. Therefore, associative hits were recorded when participants correctly identified the picture category (e.g. selecting 'object' for an object associate), while associative misses occurred when participants selected the incorrect category. Hits and misses in this task could indicate either correct/incorrect retrieval of the associated category or recognition of the word as old along with a correct/incorrect category guess. Neural measures of memory reactivation are crucial in distinguishing between 2AFC associative hits based on strong, moderate, or minimal reactivation. Average rates of associative hits and misses were calculated based on correct/incorrect responses relative to the total number of trials during the Day 2 Memory Cueing task.

## Imaging methods fMRI acquisition and preprocessing

Functional imaging data were acquired using a 3T Magnetom Prisma MRI scanner (Siemens, Germany), equipped with a 64-channel head coil. Gradient-echo T2*-weighted echoplanar images (EPIs) were acquired for functional volumes. The imaging parameters included a slice thickness of 2 mm and an isotropic voxel size of 2 mm². Sixty-two slices were aligned to the anterior commissure–posterior commissure line using a descending interleaved multiband method. The repetition time (TR) was 2000 ms, the echo time (TE) was 30 ms, the flip angle was 60°, and the field of view was 224 × 224 mm. Before the Day 2 Memory Cueing task, high-resolution T1-weighted structural images were acquired for each participant using a magnetization-prepared rapid acquisition gradient echo (MPRAGE) sequence. The structural images had a voxel size of 0.8 × 0.8 × 0.9 mm and consisted of 256 slices. The imaging parameters for the MPRAGE sequence were a TR of 2.5 s and a TE of 2.12 ms. The structural and functional images underwent preprocessing using SPM12 (http://www.fil.ion.ucl.ac.uk/spm/) implemented in MATLAB. The first three functional images of each run were discarded to avoid T1 saturation effects. Preprocessing steps included spatial realignment, slice time

correction, coregistration to the structural image, normalization to the Montreal Neurological Institute (MNI) standard space, and spatial smoothing with a 6 mm full-width at half-maximum (FWHM) Gaussian kernel.

## fMRI wholebrain GLM analysis of cued recall on days 1, 2, and 3

For each participant, a GLM was estimated using smoothed and normalized functional images for all tasks, applying a high-pass cut-off filter at 128 s to eliminate low-frequency drifts. T-statistic maps from GLM analyses represented contrasts of interest. Cluster correction via Gaussian random fields (GRF) theory corrected for multiple comparisons with a significance threshold of p>0.05. The GLM included regressors for cued recalls on Days 1 and 3: associative category hit$_{Cued\ and\ correct}$, associative miss$_{cued\ and\ correct}$, associative category hit$_{Uncued}$, and associative miss$_{Uncued}$. Trials that were 'Uncued' on Day 2 were considered not reactivated, 'Cued and correct' trials on Day 2 were considered reactivated, and trials that were cued on Day 2 but not remembered were removed from the analysis. Additionally, six regressors addressed movement realignment parameters (two run-specific and one session-specific regressor for each day). For the Memory Cueing task, regressors covered associative category hits, associative misses, six-movement realignment parameters, and one for the session, resulting in 35 regressors in total. Before the group analyses of the cued recall data, we subtracted estimates of associative missed trials from associative category hit trials in first-level estimations. Group-level analyses used a two-factorial model (Group: YOH vs. CORT vs. PLAC; Cued: correct vs. incorrect on Day 2) to examine a *Group × Reactivation* interaction. Day 2 group-level analyses employed two-sample unpaired t-tests for participant-level contrasts. The Memory Cueing task on Day 2 preceded the pharmacological manipulation, identifying Regions of Interest (ROIs) more active during associative category hits compared to associative miss during reactivation, independent of Group. A flexible factorial model based on three factors (Group, Reactivation, Day) explored group-level changes in neural activity from Day 1 to Day 3.

## ROI analyses

We examined task-evoked activation in the hippocampus and VTC, based on their central role in the domain of episodic memory retrieval (*Ranganath et al., 2004*; *Kim, 2010*), utilizing ROI masks derived from the Harvard-Oxford cortical and subcortical atlas with a 50% probability threshold. The VTC mask combined relevant regions from the Harvard-Oxford Atlas, excluding the hippocampus. In overall GLMs, the same regressors were used, but voxels were masked by a given ROI, and ROI-specific effects were small-volume corrected.

For native-space single-trial analyses, ROI masks were back-transformed using the inverse deformation field from segmentation during preprocessing. In all ROI analyses on voxel-wise modeled data, single-trial beta estimates were calculated for all days and tasks to provide a detailed characterization of memory-related neural responses. A 128 s high-pass cut-off filter removed low-frequency drifts. The models, following the 'Least-squares all' approach, were performed on realigned, slice-time corrected, native space images for subsequent multivariate pattern analyses (MVPA, RSA).

## Multivariate pattern classification

Multivariate/voxel pattern analyses (MVPA) using The Decoding Toolbox (*Hebart et al., 2015*) functions assessed trial-wise cortical reinstatement strength. In total, three L2-penalized logistic regression models (C=0.1) were employed. The first model served to evaluate the classification performance within the localizer task by utilizing leave one-run-out cross-validation (scenes vs. objects) to validate the overall quality of the task and associated data. The second model evaluated the classification performance within the localizer task by utilizing leave one-run-out cross-validation (scenes vs. objects) to validate the overall quality of the task and associated data. Model performance was assessed using classification accuracy. The third model was trained on neural patterns from the visual localizer task and served to classify remembered scenes from remembered objects, serving as the category pattern reinstatement index in further analyses. Trial-wise category pattern reinstatement evidence was assessed using logits and balanced classification accuracy, which accounts for a potentially unequal number of samples during testing.

## Tracking online reactivation

To comprehensively assess trial-wise reactivation on Day 2, we utilized reaction times, trial-wise univariate beta activity in the hippocampus and VTC, category pattern reinstatement indexed via MVPA in the VTC, and Hippocampal pattern reactivation from encoding to reactivation (Encoding-Reactivation-Similarity via RSA). Linear mixed models were employed to predict single-trial beta activity of the hippocampus and VTC, as well as category pattern reinstatement, using trial-specific Day 2 reaction times. A linear mixed model was also fit to univariate hippocampal activity predicted by category pattern reinstatement, aligning with previous findings that showed a positive association between hippocampal activity and VTC pattern reinstatement (*Gagnon et al., 2019*). The category pattern reinstatement index and hippocampal pattern reactivation were used to classify trials as 'high' or 'low' online reactivation, predicting Day 3 performance in GLMMs with information from all available trials.

## Representational similarity analyses

To assess drug- and reactivation-related changes in Day 3 neural patterns between cued and correct and uncued trials, we conducted a Representational Similarity Analysis (RSA), focusing on the hippocampus using customized scripts from The Decoding Toolbox (*Polyn et al., 2009*). Beta vectors from single-trial GLMs were extracted, and RSA was conducted in the native space using participant-specific hippocampal masks. The representational similarity (Fisher z-transformed) from Day 1 encoding (average across three encoding runs) to Day 2 reactivation ('Day 1 Day 2 encoding-reactivation similarity (ERS) analysis') captured trial-specific pattern changes, which were assumed to provide a measure of neural memory reactivation and were used to predict Day 3 memory performance in GLMMs on a trial-by-trial basis.

## Statistical analyses

Univariate fMRI statistical tests were conducted in the SPM12 environment (http://www.fil.ion.ucl.ac.uk/spm/). All other statistical models and tests were conducted in the R environment (version 3.3.4). Reported p-values resulting from ANOVAs were Greenhouse-Geisser corrected, when required; univariate fMRI voxel-cluster results were FWE corrected. Baseline and control variables on Days 1 and 3 (e.g. blood pressure) were tested with one-way ANOVAs. Day 2 parameters validating the effective pharmacological manipulation (i.e. blood pressure, heart rate, mood, cortisol, SCR) were tested with repeated-measures ANOVAs (within-subject factor Time, between-subject factor Group) and subsequent post-hoc t-tests. All reported p-values from statistical models and post-hoc tests reported throughout the manuscript were corrected for multiple comparisons with Bonferroni correction (for a number of ROIs and tests, respectively). Measures of task performance, including hits, false alarms, and d', that investigated the pharmacological effect on later memory for reactivated trials were subjected to repeated-measures ANOVAs (within-subject factor Reactivated, between-subject factor Group) and subsequent post-hoc t-tests. For calculations of associative d', values of zero were replaced with 0.5/denominator and values of 1 with 1–0.5/denominator (*Macmillan and Kaplan, 1985*).

Single-trial analyses were modeled using (Generalized) Linear Mixed Models predicting associative category hits/misses on Day 3, based upon several different predictor variables (i.e. Reactivation, Group, Reaction times). Reaction times serve as a proxy for memory confidence and memory strength, with faster responses reflecting higher confidence/strength and slower responses suggesting greater uncertainty/weaker memory. The association between reaction times and memory confidence has been established by previous research (*Starns, 2021*; *Robinson et al., 1997*; *Ratcliff and Murdock, 1976*), suggesting that the distinction between high from low confidence responses differentiates vividly recalled associations from decisions based on weaker memory evidence. Reaction times are further linked to hippocampal activity during recall tasks (*Heinbockel et al., 2024*; *Johnson and Rugg, 2007*), and stress effects on memory are particularly pronounced for high-confidence memories (*Johnson and Rugg, 2007*). We conducted a median-split within each participant to categorize trials as slow vs. fast reaction time trials during Day 2 memory cueing. We chose to conduct this split on the participant- and not group level because there is substantial inter-individual variability in overall reaction times and to retain an equal number of trials in the low and high confidence conditions. Models were fitted with the lme4 (*Bates et al., 2014*) statistical package. Models were estimated using a restricted maximum likelihood (REML) approach. The Resulting p-values were Bonferroni corrected

for the number of ROIs. Post-hoc slope comparisons of GLMMs were conducted using the *emtrends* (*Searle et al., 1980*) function including Tukey correction. Visualization and analysis utilized the R package ggplot2 (*Wickham, 2024*) as well as Inkscape (https://inkscape.org).

## Acknowledgements

We gratefully acknowledge the support of Mali Wichmann, Ann-Kathrin Otte, and Flavia Holzki during recruiting, and data acquisition. German Research Foundation (DFG) grant as part of the collaborative research centre 936 'Multisite Communication in the Brain' (SFB 936/B10; Project Nr. 178316478) to LS.

## Additional information

### Funding

| Funder | Grant reference number | Author |
| --- | --- | --- |
| Deutsche Forschungsgemeinschaft | 178316478 | Lars Schwabe |
| Deutsche Forschungsgemeinschaft | SFB 936/B10 | Lars Schwabe |

The funders had no role in study design, data collection and interpretation, or the decision to submit the work for publication.

### Author contributions

Hendrik Heinbockel, Data curation, Formal analysis, Validation, Visualization, Methodology, Writing – original draft, Writing – review and editing; Gregor Leicht, Resources, Writing – review and editing; Anthony D Wagner, Conceptualization, Writing – review and editing; Lars Schwabe, Conceptualization, Resources, Supervision, Funding acquisition, Writing – original draft, Project administration, Writing – review and editing

### Author ORCIDs

Hendrik Heinbockel ⓘ https://orcid.org/0000-0002-9149-6755
Gregor Leicht ⓘ http://orcid.org/0000-0002-5104-9336
Anthony D Wagner ⓘ https://orcid.org/0000-0003-0624-4543
Lars Schwabe ⓘ https://orcid.org/0000-0003-4429-4373

### Ethics

Human subjects: All participants provided written informed consent before the start of the experiment and received a monetary compensation for their participation. The study was approved by the ethics committee of the Medical Chamber of Hamburg (PV5960).

Reviewer #1 (Public review): https://doi.org/10.7554/eLife.100525.3.sa1
Reviewer #2 (Public review): https://doi.org/10.7554/eLife.100525.3.sa2
Author response https://doi.org/10.7554/eLife.100525.3.sa3

## Additional files

### Supplementary files

MDAR checklist

### Data availability

All behavioural and (anonymized) functional MRI data as well as analysis scripts have been deposited and are publicly available at https://doi.org/10.25592/uhhfdm.14137. Any additional information required to re-analyse the data reported in this paper as well as raw and native space (not de-faced) MRI images are available from the lead contact upon request.

The following dataset was generated:

| Author(s) | Year | Dataset title | Dataset URL | Database and Identifier |
|-----------|------|---------------|-------------|------------------------|
| Heinbockel H, Wagner AD, Schwabe L, Leicht G | 2025 | Post-retrieval noradrenergic activation impairs subsequent memory depending on cortico-hippocampal reactivation | https://doi.org/10.25592/uhhfdm.14137 | Research Data Repository (FDR@UHH), 10.25592/uhhfdm.14137 |

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

## Appendix 1

### Control variables

We controlled for potential group differences in depressive mood, chronic stress as well as state and trait anxiety. Importantly, the three groups did not differ in any of these variables (depressive mood: $F(1,60) = 0.67$, p=0.406, $\eta^2$=0.01, state anxiety: $F(1,60) = 0.31$, p=0.581, $\eta^2$<0.01; trait anxiety: $F(1,60) = 1.07$, p=0.305, $\eta^2$=0.02, chronic stress: $F(1,60) = 0.35$, p=0.557, $\eta^2$<0.01). See *Appendix 1—table 6*.

### Inter-individual differences in pharmacological responses did not significantly mediate subsequent memory

To account for individual differences in cortisol responses after pill intake, we fit additional GLMMs predicting Day 3 subsequent memory of cued and correct trials including the factors of *Individual baseline-to-peak cortisol* and *Group*. Doing so allowed us to account for variation in Day 3 performance, which might have resulted from within-group variation in cortisol responses, in particular in the CORT group. Importantly, none of the models predicting Day 3 memory performance by Day 2 cortisol-increase and Group, median-split RTs (high/low), hippocampal activity and RTs, or hippocampal activity and VTC category reinstatement revealed a significant group x baseline-to-peak cortisol interaction (all Ps>0.122). These results suggest that inter-individual differences in cortisol responses did not have a significant impact on subsequent memory, beyond the influence of group per se. The same analyses were repeated for systolic blood pressure employing GLMMs predicting Day 3 subsequent memory of cued and correct trials including the factors of *Individual baseline-to-peak systolic blood pressure* and *Group* to account for variation in Day 3 performance, which might have resulted from within-group variation in blood pressure response, in particular in the YOH group. While the model predicting Day 3 memory performance revealed a significant *Individual baseline-to-peak systolic blood pressure × Group × median-split RTs (high/low)* interaction ($\beta$=–0.05 ± 0.02, z=–2.04, p=0.041, $R^2_{conditional}$ = 0.01), post-hoc slope tests, however, did not show any significant difference between groups (all $P_{Corr}$ >0.329). The remaining models including hippocampal activity and RTs, or hippocampal activity and VTC category reinstatement did not reveal a significant Group ×*Individual baseline-to-peak systolic blood pressure* interaction (all Ps>0.101). These results suggest that inter-individual differences in systolic blood pressure responses did not have a significant impact on subsequent memory, beyond the influence of group per se.

### Offline encoding and cueing reinstatement in hippocampus and cortical areas

Aside from examining neural activity related to retrieval during the Memory Cueing task, we also investigated evidence of offline reactivation, which is manifested in neural reinstatement observed during the resting-state scan conducted both before and after the Memory Cueing task. An ANOVA including the factors *Group*, *Time* (pre, post), and *Threshold* (pre, post) revealed a significant Time × *Threshold* interaction in the hippocampus ($F(1,58) = 40.48$, p<0.001, $\eta^2$=0.02). Post-hoc t-tests confirmed a significant increase from pre- to post-cueing hippocampal offline reactivation in the pre-threshold condition (post-pre: t(60) = –3.21, p=0.006, d=0.59) compared to post-threshold (post-pre: t(60) = –0.78, p=1, d=0.05), without a difference between groups (all Ps>0.162). In addition to this analysis of the offline reactivation of the memory cueing events, we analyzed the hippocampal offline reactivation of encoding patterns. We again utilized an ANOVA including the factors *Group*, *Time* (pre, post), and *Threshold* (pre, post) which did not reveal any significant main effect or interaction of the factors (all Ps>0.823) indicating that patterns from Day 1 memory encoding were not reactivated offline in the hippocampus after Day 2, either before memory cueing or after reactivation during memory cueing. We did however not observe a difference in post-cueing offline reactivation of encoding vs. memory cueing patterns (p=0.393). Thus, while we obtained evidence for hippocampal offline reactivation of the Day 2 cueing events, there was no significant evidence for hippocampal offline reactivation of the encoded representation after Day 2 Memory Cueing.

Next, we tested for offline reactivation of patterns from memory cueing in the post- vs. pre-cueing resting state intervals in VTC. An ANOVA including the factors *Group*, *Time* (pre, post), and *Threshold* (pre, post) revealed a significant Time × *Threshold* interaction ($F(1,58) = 49.42$, p<0.001, $\eta^2$=0.02). Post-hoc t-test confirmed again a significant increase from pre- to post-cueing offline reactivation in the pre-threshold condition (post-pre: t(60) = –2.84, p=0.018, d=0.51) compared to post-threshold condition (post-pre: t(60) = 0.30, p=1, d=0.05). Testing for offline reinstatement of

Day 1 encoding patterns revealed a similar pattern of results. Again, a significant Time × *Threshold* interaction was observed (F(1,58) = 32.63, p<0.001, $\eta^2$=0.02), indicating generally stronger reactivation of encoding patterns after the memory cueing task. Notably, however, we also observed significant interactions including the factor *Group* (Group × *Time*: F(2,58) = 5.58, p=0.009, $\eta^2$=0.08; Group × *Threshold*: F(2,58) = 6.61, p=0.018, $\eta^2$=0.09). Post-hoc t-tests revealed that in the post-cueing interval, the PLAC group showed significantly fewer reactivation events in the VTC compared to the YOH and CORT groups (PLAC vs. YOH: t(28.35)=−2.82, p=0.017, d=0.86; PLAC vs. CORT: t(35.92)=−3.09, p=0.007, d=0.96). We did not observe a difference in post-cueing offline reactivation of encoding vs. memory cueing patterns (p=0.393). For the VTC, we obtained evidence suggesting offline reactivation of the Day 2 cueing events as well as the encoded representations following Day 2 Memory Cueing.

Finally, we tested for offline reactivation of patterns from memory cueing in the post- vs. pre-cueing resting state intervals in PCC. An ANOVA including the factors *Group*, *Time* (pre, post) and *Threshold* (pre, post) revealed a significant Time × *Threshold* interaction (F(1,58) = 22.94, p<0.001, $\eta^2$=0.01). A Post-hoc t-test revealed a trend for a difference between pre-and post-cueing offline reactivation in the pre-threshold condition (post-pre: t(60) = −2.80, p=0.057, d=0.43), and no difference in the post-threshold condition (post-pre: t(60) = −0.13, p=1, d=0.02). Offline reactivation analysis of encoding patterns revealed a similar pattern of results. Again, a significant Time × *Threshold* interaction was observed (F(1,58) = 63.13, p<0.001, $\eta^2$=0.01), indicating generally more reactivation of Day 1 encoding patterns after the memory cueing task. Post-hoc t-tests revealed a trend-level difference between pre-and post-cueing offline reactivation in the pre-threshold condition (post-pre: t(60) = −2.41, p=0.056, d=0.44), and no significant difference in the post-threshold condition (t(60) = 0.25, p=1, d=0.05). While we again found evidence indicating offline reactivation of the Day 2 cueing events in PCC, there was only trend-level evidence supporting PCC offline reactivation of the encoding representations following Day 2 Memory Cueing.

## Day 3 offline pattern reactivation does not modulate effects of post-retrieval noradrenergic activation on subsequent memory performance

Next, we asked whether offline reactivation of the retrieval patterns during Day 2 memory cueing interacted with the effects of post-retrieval YOH and CORT in light of subsequent memory effects. We classified cued and correct (Day 2) trials as strongly or weakly reactivated based on a median split on region-specific (hippocampus, VTC, PCC) offline pattern reinstatement, allowing us to include the uncued trials in further analyses. We employed a GLMM with *Offline reactivation* (post-pre) and *Group* as predictors of Day 3 associative category hits. This analysis did not yield any main effect or interaction including the factor Group (all Ps>0.215) in our ROIs (hippocampus, VTC, PCC), suggesting that offline processing of the memory cueing (retrieval-related) patterns was not affected by the pharmacological manipulation. In the next step, we included the offline reinstatement from the encoding task (Day 1) in the model, predicting Day 3 associative category hits. This GLMM did not reveal any main effect or interaction including the factor Group either (all Ps>.993). Together, Day 2 offline reinstatement of encoding- or retrieval-related patterns did not modulate the impact of the post-retrieval stress system manipulations on subsequent memory.

**Appendix 1—table 1.** Memory performance and reaction times across experimental days.

| | PLAC | YOH | CORT |
|---|---|---|---|
| Day 1 (cued and correct on Day 2) | | | |
| Hits (%) | 49.72 (1.64) | 43.19 (2.15) | 49.59 (2.82) |
| Dprime | 1.21 (0.07) | 1.01 (0.07) | 1.25 (0.11) |
| Reaction time (s) | 2.29 (0.07) | 2.40 (0.07) | 2.39 (0.07) |
| Day 1 (not cued on Day 2) | | | |
| Hits (%) | 49.38 (1.96) | 41.77 (2.11) | 50.64 (2.64) |

*Appendix 1—table 1 Continued on next page*

*Appendix 1—table 1 Continued*

|  | PLAC | YOH | CORT |
|---|---|---|---|
| Dprime | 1.19 (0.09) | 0.96 (0.07) | 1.17 (0.10) |
| Reaction time (s) | 2.27 (0.07) | 2.41 (0.09) | 2.44 (0.07) |
| | | | |
| Day 2 (cued and correct on Day 2) | | | |
| Hits (%) | 68.85 (1.52) | 63.95 (8.56) | 69.70 (13.34) |
| Reaction time | 2.27 (0.63) | 2.31 (0.65) | 2.24 (0.63) |
| Reaction time (fast; s) | 1.86 (0.05) | 1.92 (0.05) | 1.82 (0.04) |
| Reaction time (slow; s) | 2.80 (0.06) | 2.89 (0.07) | 2.80 (0.06) |
| | | | |
| Day 3 (cued and correct on Day 2) | | | |
| Hits (%) | 74.32 (1.68) | 65.24 (2.12) | 69.80 (2.52) |
| Dprime | 2.25 (0.11) | 1.78 (0.10) | 2.14 (0.13) |
| Reaction time (s) | 1.98 (0.08) | 2.04 (0.07) | 1.98 (0.07) |
| | | | |
| Day 3 (not cued on Day 2) | | | |
| Hits (%) | 46.71 (1.78) | 36.25 (1.80) | 46.16 (2.39) |
| Dprime | 1.16 (0.09) | 0.74 (0.07) | 1.09 (0.09) |
| Reaction time (s) | 2.29 (0.07) | 2.30 (0.08) | 2.31 (0.07) |

Reaction times relate hits specifically. Data represent means ± SE.

**Appendix 1—table 2.** Physiological parameters and mood at baseline across Day 1 and Day 3.

|  |  | PLAC | YOH | CORT |
|---|---|---|---|---|
| Day 1 | Heart rate (bpm) | 78.85 (2.67) | 75.97 (2.77) | 76.35 (2.52) |
|  | Systolic blood pressure (mmHg) | 107.80 (2.80) | 116.47 (2.63) | 114.66 (3.66) |
|  | Diastolic blood pressure (mmHg) | 68.32 (1.87) | 71.78 (2.18) | 71.00 (2.29) |
|  | cortisol (nmol) | 9.49 (2.02) | 7.16 (1.16) | 10.76 (1.56) |
|  | Mood (good/bad) | 34.80 (0.90) | 33.47 (0.98) | 33.70 (0.87) |
|  | Tiredness (energized/tired) | 31.05 (1.06) | 30.76 (1.14) | 32.30 (1.01) |
|  | Calmity (calm/restless) | 30.50 (1.15) | 30.95 (1.47) | 28.90 (1.07) |
|  |  |  |  |  |
| Day 3 | Heart rate (bpm) | 78.65 (2.46) | 78.28 (2.31) | 78.26 (2.38) |
|  | Systolic blood pressure (mmHg) | 112.22 (2.14) | 112.42 (3.26) | 114.04 (3.86) |
|  | Diastolic blood pressure (mmHg) | 72.15 (1.48) | 72.66 (1.54) | 68.83 (1.95) |
|  | cortisol (nmol) | 8.41 (1.59) | 5.88 (1.12) | 7.32 (1.01) |
|  | Mood (good/bad) | 34.30 (1.05) | 34.71 (0.81) | 34.10 (0.91) |

*Appendix 1—table 2 Continued on next page*

*Appendix 1—table 2 Continued*

|  | | PLAC | YOH | CORT |
|---|---|---|---|---|
|  | Tiredness (energized/tired) | 33.40 (1.05) | 34.09 (0.98) | 32.80 (0.85) |
|  | Calmity (calm/restless) | 30.50 (1.15) | 31.00 (1.20) | 30.90 (1.00) |

Subjective and physiological parameters of participants taken at the beginning of Day 1 and Day 2. There were no significant difference in either subjective or physiological stress parameters between groups. Data represent means (±SE).

**Appendix 1—table 3.** Significant clusters in the whole-brain analyses of Day 2 memory cueing (correct – incorrect).

| Region | Central coordinates (x, y, z; MNI) | Cluster-P (FWE 0.05) | Cluster-T |
|---|---|---|---|
| Frontal Medial, ACC | –4, 50, –8 | <0.001 | 10.72 |
| Lateral Occipital L, AG L | –48, –66, 30 | <0.001 | 9.08 |
| Amygdala L, Striatum L | –14, 4, –16 | <0.001 | 8.22 |
| Posterior Cingulate Cortex | 4, –41, 38 | <0.001 | 8.10 |
| Supramarginal Gyrus R | 62, –44, 20 | <0.001 | 7.94 |
| hippocampus L | –26, –32, –10 | <0.001 | 7.93 |
| hippocampus R | 32, –40, –12 | <0.001 | 7.89 |
| Frontal Pole | –22, 38, 42 | <0.001 | 7.32 |
| Temporooccipital Cortex R | 52, –50, –14 | <0.001 | 6.13 |
| Superior Parietal R | 30, –46, 62 | <0.001 | 6.11 |
| Temporooccipital Cortex L | –52, –62, –2 | <0.001 | 6.93 |
| Postcentral Gyrus | 50, –22, 50 | <0.001 | 5.77 |
| Superior Frontal Gyrus | –20, 2, 56 | <0.001 | 6.67 |
| Putamen L | –26,–8, 8 | <0.001 | 6.64 |
| Mid temporal gyrus L | –62,–16, –8 | <0.001 | 6.37 |

Depicted clusters are ordered by cluster-peak T-values.

**Appendix 1—table 4.** Results from Day 2 gPPI analysis (remembered – forgotten).

| Seed | Target | Cluster size | pFWE (cluster) | pFWE (peak) |
|---|---|---|---|---|
| PCC | Lat. OCC left | 237 | <0.001 | 0.015 |
|  | VTC | 1218 | <0.001 | <0.001 |
| MPFC | Lat. OCC left | 404 | <0.001 | <0.001 |
|  | Sup. Parietal lobe | 836 | <0.001 | <0.001 |
|  | VTC | 1168 | <0.001 | 0.002 |
| Left Hippocampus | VTC | 260 | <0.001 | 0.016 |
| VTC | Lat. OCC left | 789 | <0.001 | <0.001 |
|  | Sup. Parietal lobe | 1123 | <0.001 | <0.001 |
|  | VTC | 260 | <0.001 | 0.016 |

*Appendix 1—table 4 Continued on next page*

*Appendix 1—table 4 Continued*

| Seed | Target | Cluster size | pFWE (cluster) | pFWE (peak) |
|------|--------|--------------|----------------|-------------|
| PCC | Lat. OCC left | 237 | <0.001 | 0.015 |
| | VTC | 1218 | <0.001 | <0.001 |
| MPFC | Lat. OCC left | 404 | <0.001 | <0.001 |
| | Sup. Parietal lobe | 836 | <0.001 | <0.001 |
| | VTC | 1168 | <0.001 | 0.002 |
| Left Hippocampus | VTC | 260 | <0.001 | 0.016 |
| VTC | Lat. OCC left | 789 | <0.0001 | <0.001 |
| | Sup. Parietal lobe | 1123 | <0.001 | <0.001 |

All p-values are Bonferroni-corrected for amount of ROIs. Only clusters with a minimum of 25 voxels were included.

**Appendix 1—table 5.** Physiological parameters across Day 2.

| | | PLAC | YOH | CORT |
|------|------|------|------|------|
| Heart rate (bpm) | Base | 80.42 (2.41) | 75.02 (2.10) | 77.52 (1.97) |
| | Task | 87.40 (2.29) | 81.61 (2.06) | 82.76 (1.86) |
| | Resting-state post | 80.15 (2.28) | 74.42 (2.21) | 75.28 (2.18) |
| | +40 | 70.35 (2.58) | 64.92 (2.06) | 67.04 (2.10) |
| | +55 | 70.65 (2.28) | 65.59 (1.97) | 67.52 (2.33) |
| | +70 | 69.12 (2.18) | 67.45 (2.37) | 65.61 (2.03) |
| | +85 | 68.90 (2.40) | 67.47 (1.97) | 65.73 (2.12) |
| | +100 | 70.25 (2.78) | 69.45 (2.60) | 67.85 (2.16) |
| Systolic BP (mmHg) | Base | 113.00 (3.11) | 115.42 (3.11) | 117.95 (3.23) |
| | +40 | 115.35 (1.95) | 118.00 (3.27) | 117.38 (3.70) |
| | +55 | 110.40 (1.75) | 118.61 (3.36) | 115.83 (3.11) |
| | +70 | 108.45 (1.71) | 120.92 (3.43) | 115.11 (3.68) |
| | +85 | 108.50 (2.27) | 122.38 (3.57) | 115.28 (3.34) |
| | +100 | 107.55 (2.46) | 123.69 (3.29) | 115.78 (3.21) |
| Diastolic BP (mmHg) | Base | 72.53 (1.99) | 71.86 (2.61) | 74.57 (2.14) |
| | +40 | 73.22 (2.26) | 74.14 (1.96) | 73.76 (2.08) |
| | +55 | 72.33 (1.45) | 71.02 (2.10) | 72.67 (2.05) |
| | +70 | 71.28 (1.64) | 73.31 (2.36) | 72.98 (2.18) |
| | +85 | 72.85 (1.67) | 75.79 (2.12) | 73.31 (2.30) |
| | +100 | 71.55 (1.71) | 77.29 (2.30) | 73.98 (1.81) |

*Appendix 1—table 5 Continued*

|  |  | PLAC | YOH | CORT |
|---|---|---|---|---|
| cortisol (nmol) | Base | 11.14 (2.14) | 7.22 (1.17) | 8.18 (1.13) |
|  | +20 | 5.40 (0.78) | 4.04 (0.39) | 8.15 (2.03) |
|  | +40 | 4.53 (0.62) | 2.92 (0.31) | 24.59 (6.32) |
|  | +70 | 3.63 (0.60) | 3.14 (0.41) | 57.60 (4.88) |
|  | 100 | 2.70 (0.34) | 4.05 (0.69) | 52.93 (4.95) |
| Skin conductance | Resting-state pre | 5.82 (1.35) | 3.32 (0.55) | 5.97 (0.86) |
|  | Task | 5.88 (1.47) | 3.07 (0.75) | 6.71 (1.31) |
|  | Resting-state post | 9.28 (1.63) | 6.53 (0.96) | 10.02 (1.25) |

Physiological parameters of participants across Day 2 relative to drug administration. Systolic blood pressure significantly increased over time in the YOH group, compared to the CORT and PLAC groups. Salivary cortisol increased in the CORT group but not in the YOH or PLAC group. Data represent means (±SE).

**Appendix 1—table 6.** Subjective mood scores across Day 2.

|  |  | PLAC | YOH | CORT |
|---|---|---|---|---|
| MDBF *Base* | Mood (good/bad) | 34.85 (1.11) | 34.38 (0.78) | 34.90 (0.72) |
|  | Tiredness (energized/tired) | 33.85 (1.10) | 33.33 (1.04) | 33.70 (1.03) |
|  | Calmness (calm/restless) | 34.85 (1.11) | 34.38 (0.78) | 34.90 (0.72) |
| MDBF *+40* | Mood (good/bad) | 35.05 (1.08) | 34.81 (0.81) | 33.70 (0.81) |
|  | Tiredness (energized/tired) | 33.70 (0.94) | 33.53 (1.06) | 32.95 (1.01) |
|  | Calmness (calm/restless) | 30.55 (0.87) | 28.90 (1.47) | 30.04 (0.91) |
| MDBF *+55* | Mood (good/bad) | 34.70 (0.86) | 35.14 (0.80) | 33.20 (1.09) |
|  | Tiredness (energized/tired) | 34.10 (0.98) | 33.80 (0.92) | 32.90 (1.10) |
|  | Calmness (calm/restless) | 30.00 (0.97) | 30.47 (1.30) | 28.45 (1.75) |
| MDBF *+70* | Mood (good/bad) | 34.55 (1.11) | 34.81 (0.86) | 32.75 (1.07) |
|  | Tiredness (energized/tired) | 33.65 (1.08) | 34.28 (1.08) | 32.75 (1.16) |
|  | Calmness (calm/restless) | 29.55 (1.43) | 30.81 (1.34) | 29.60 (1.38) |
| MDBF *+85* | Mood (good/bad) | 34.20 (1.14) | 34.09 (0.93) | 32.30 (1.20) |
|  | Tiredness (energized/tired) | 33.20 (1.18) | 33.47 (1.23) | 31.60 (1.14) |
|  | Calmness (calm/restless) | 28.80 (1.73) | 30.71 (1.56) | 28.55 (1.58) |
| MDBF *+100* | Mood (good/bad) | 33.30 (1.39) | 33.19 (1.16) | 32.55 (1.02) |
|  | Tiredness (energized/tired) | 32.70 (1.34) | 32.61 (1.38) | 32.50 (1.26) |

*Appendix 1—table 6 Continued on next page*

*Appendix 1—table 6 Continued*

| | PLAC | YOH | CORT |
|---|---|---|---|
| Calmness (calm/restless) | 30.20 (1.36) | 29.19 (1.56) | 29.40 (1.48) |

Subjective mood ratings according to the *Mehrdimensionale Befindlichkeitsfragebogen* (MDBF **Katsuki et al., 1997**) across Day 2. Scores did not reveal a significant difference in either scale between groups. Data represent means (±SE).

**Appendix 1—table 7.** Participants' state, trait anxiety, chronic stress, and depression scores.

| | PLAC | YOH | CORT |
|---|---|---|---|
| Depression score | 6.20 (0.89) | 9.04 (1.20) | 7.47 (1.00) |
| State anxiety | 43.20 (1.02) | 42.09 (0.94) | 42.38 (1.16) |
| Trait anxiety | 43.05 (1.41) | 46.95 (1.79) | 45.19 (0.93) |
| Chronic stress | 70.95 (6.86) | 90.61 (8.42) | 77.14 (5.37) |

State and Trait anxiety scores were measured with the State-Trait Anxiety Inventory. Depression Scores were determined utilizing the Beck Depression Inventory. Chronic stress was measured with the Trier Inventory of Chronic Stress. Participants conducted the three questionnaires at home before the actual experiment started. Data represent means (±SE).

## Appendix 2

### Stimulus material

The words of each associative word-picture pair had either negative (mean valence = 3.45, mean arousal = 5.72, mean concreteness = 4.62) or neutral valence (mean valence = 5.06, mean arousal = 2.15, mean concreteness = 4.41). These words were selected from the Leipzig Affective Norms for German *Strange et al., 2014*. Since there was no significant influence of word valence at the behavioural and neural levels, which may be due to the fact that the arousal evoked by emotional words is typically significantly lower than for pictures or movies, we did not include the factor valence in the analyses reported here.

### Catch trials

Out of the 164 word-picture pairs presented during encoding, 20 pairs were designated as catch trials for the subsequent cued recall tasks. The selection of word-picture catch trial pairs was counterbalanced in terms of valence (negative/neutral) and category (scene/object). Catch trials served to maintain participants' attention during the cued recall tests and to motivate participants to retrieve the associated picture while seeing the associated word. To further motivate participants to reactivate the associated picture in as much detail as possible when seeing the word cue, participants were informed that correctly answered catch trials would increase their financial compensation. The cued recall tests on Days 1 and 3 included eight catch trials each, while the shorter Day 2 Memory Cueing task included four catch trials. The temporal position of catch trials was distributed within a task, ensuring equal spacing between them. A catch trial was triggered when participants correctly designated the presented word as 'old,' 'old/scene' or 'old/object.' Upon this choice, either the corresponding or a semantically similar picture probe was displayed on the screen for 0.5 s and participants had to judge whether the probe was the studied associate of the word, responding 'yes' or 'no' within 1 s. Catch trial performance did not differ between groups on any experimental day (all Ps >0.603). All catch trials were subsequently excluded from the analyses to prevent potential biases in memory effects due to the re-presentation of correct or semantically similar picture probes together with old words.

### Tracking offline reactivation

In addition to retrieval-related neural activity during the Memory Cueing task, we also tested for evidence of offline reactivation, reflected in neural reinstatement during the resting-state scan before and after the Memory Cueing task. To do so, we employed an RSA approach previously utilized to analyse offline replay *Kausche et al., 2021*, correlating trial/event-specific neural patterns from the Memory Cueing task (Day 2) with patterns from the pre-and post-memory cueing resting-state fMRI scans, separately for the hippocampus, VTC, and PCC. To compare the number of pre- and post-reactivation events, we initially calculated the mean + 1.5 SDs across all correlations within the pre-cueing phase (*pre-threshold*). This threshold was then applied to pre-and post-cueing correlation matrices. To validate potential effects, we repeated this approach but applied the threshold (mean + 1.5 SDs) from the post-cueing interval (*post-threshold*) to the pre-and post-cueing correlation matrices. Offline reactivation events were then quantified as the absolute number of surviving correlations.

