## [Editor Report · eLife Assessment]

This work presents **important** findings of a modulatory effect of yohimbine, an alpha2-adrenergic antagonist that raises noradrenaline levels, on the reconsolidation of emotionally neutral word-picture pairs, depending on the hippocampal and cortical reactivation during retrieval. The evidence supporting the main conclusions is **convincing**, with an elegant design combining fMRI and psychopharmacology. The work will be of broad interest to researchers working on memory.

---

## [Referee Report · Reviewer #1 (Public review)]

Summary:

How reconsolidation works - particularly in humans - remains largely unknown. With an elegant, 3-day design, combining fMRI and psychopharmacology, the authors provide evidence for a certain role for noradrenaline in the reconsolidation of memory for neutral stimuli. All memory tasks were performed in the context of fMRI scanning, with additional resting state acquisitions performed before and after recall testing on Day 2. On Day 1, 3 groups of healthy participants encoded word-picture associates (with pictures being either scenes or objects) and then performed an immediate cued recall task to presentation of the word (answering is the word old or new, and was it paired with a scene or an object). On Day 2, the cued recall task was repeated using half of the stimulus set words encoded on Day 1 (only old words were presented, with subjects required to indicate prior scene vs object pairing). This test was immediately preceded by the oral administration of placebo, cortisol, or yohimibine (to raise noradrenaline levels) depending on group assignment. On Day 3, all words presented on Day 1 were presented. As expected, on Day 3, memory was significantly enhanced for associations that were cued and successfully retrieved on Day 2 compared to uncued associations. However, for associative d', there was no Cued × Group interaction nor a main effect of Group, i.e., on the standard measure of memory performance, post-retrieval drug presence on Day 2 did not affect memory reconsolidation. As further evidence for a null result, fMRI univariate analyses showed no Cued × Group interactions in whole-brain or ROI activity.

Strengths:

There are some aspects of this study that I find impressive. The study is well-designed and the fMRI analysis methodology innovative and sound. The authors have made meticulous and thorough physiological measurements, and assays of mood, throughout the experiment. By doing so, they have overcome, to a considerable extent, the difficulties inherent in timing of human oral drug delivery in reconsolidation tasks, where it is difficult to have drug present in the immediate recall period without affecting recall itself. This is beautifully shown in Fig. 3. I also think that having some neurobiological assay of memory reactivation when studying reconsolidation in humans is critical, and the authors provide this. While multi-voxel patterns of hemodynamic responses are, in my view, very difficult to equate with an "engram", these patterns do have something to do with memory.

Weaknesses:

I have major issues regarding the behavioral results and the framing of the manuscript:

(1) To arrive at group differences in memory performance, the authors performed median splitting of Day 3 trials by short and long reaction times during memory cueing on Day 2, as they took this as a putative measure of high/low levels of memory reactivation. Associative category hits on Day 3 showed a Group by Day 2 Reaction time (short, long) interaction, with post-hocs showing (according to the text) worse memory for short Day 2 RTs in the yohimbine group. These post-hocs should be corrected for multiple comparisons, as the result is not what would be predicted (see point 2). My primary issue here is that we are not given RT data for each group, nor is the median splitting procedure described in the methods. Was this across all groups, or within groups? Are short RTs in the yohimbine group any different from short RTs in the other two groups? Unfortunately, we are not given Day 2 picture category memory levels or reaction times for each group. This is relevant because (as given in Supplemental Table S1) memory performance (d´) for the Yohimbine group on Day 1 immediate testing is (roughly speaking) 20% lower than the other 2 groups (independently of whether the pairs will be presented again the following day). I appreciate that this is not significant in a group x performance ANOVA but how does this relate to later memory performance? What were the group-specific RTs on Day 1? So, before the reader goes into the fMRI results, there are questions regarding the supposed drug-induced changes in behavior. Indeed, in the discussion, there is repeated mention of subsequent memory impairment produced by yohimbine but the nature of the impairment is not clear.

This weakness was satisfactorily addressed in one revision round. As RT data are often not normally distributed, were they transformed prior to entry into linear models?

(2) The authors should be clearer as to what their original hypotheses were, and why they did the experiment. Despite being a complex literature, I would have thought the hypotheses would be reconsolidation impairment by cortisol and enhancement by yohimbine. Here it is relevant to point out that - only when the reader gets to the Methods section - there is mention of a paper published by this group in 2024. In this publication, the authors used the same study design but administered a stress manipulation after Day 2 cued recall, instead of a pharmacological one. They did not find a difference in associative hit rate between stress and control groups, but - similar to the current manuscript - reported that post-retrieval stress disrupts subsequent remembering (Day 3 performance) depending on neural memory reinstatement during reactivation (specifically driven by the hippocampus and its correlation with neocortical areas).

Instead of using these results, and other human studies, to motivate the current work, reference is made to a recent animal study: Line 169 "Building on recent findings in rodents (Khalaf et al. 2018), we hypothesized that the effects of post-retrieval noradrenergic and glucocorticoid activation would critically depend on the reinstatement of the neural event representation during retrieval". It is difficult to follow that a rodent study using contextual fear conditioning and examining single neuron activity to remote fear recall and extinction would be relevant enough to motivate a hypothesis for a human psychopharmacological study on emotionally neutral paired associates.

Minor comments

- Related to Major issue 2. In the introduction, it would be helpful to be specific about the type of memory being probed in the different studies referenced (episodic vs conditioning). For the former, please make it clear whether stimuli to be remembered were emotional or neutral, and for which stimulus class drug effects were observed. This is particularly important given that in the first paragraph you describe memory reactivation in the context of traumatic memories via mention of PTSD. It would also be helpful to know to which species you refer. For example, in line 115, "timing of drug administration..." a rodent and a human study are cited.

This weakness was addressed in one revision round, resulting in an excellent introduction, highlighting the importance of studying post-retrieval effects for memory researchers and healthcare workers.

---

## [Referee Report · Reviewer #2 (Public review)]

Summary:

The authors aimed to investigate how noradrenergic and glucocorticoid activity after retrieval influence subsequent memory recall with a 24-hour interval, by using a controlled three-day fMRI study involving pharmacological manipulation. They found that noradrenergic activity after retrieval selectively impairs subsequent memory recall, depending on hippocampal and cortical reactivation during retrieval.

Overall, there are several significant strengths for this well-written manuscript.

---

## [Author Response]

The following is the authors’ response to the original reviews.

**Reviewer #1 (Public review):**
Summary:How reconsolidation works - particularly in humans - remains largely unknown. With an elegant, 3-day design, combining fMRI and psychopharmacology, the authors provide evidence for a certain role for noradrenaline in the reconsolidation of memory for neutral stimuli. All memory tasks were performed in the context of fMRI scanning, with additional resting-state acquisitions performed before and after recall testing on Day 2. On Day 1, 3 groups of healthy participants encoded word-picture associates (with pictures being either scenes or objects) and then performed an immediate cued recall task to presentation of the word (answering is the word old or new, and whether it was paired with a scene or an object). On Day 2, the cued recall task was repeated using half of the stimulus set words encoded on Day 1 (only old words were presented, with subjects required to indicate prior scene vs object pairing). This test was immediately preceded by the oral administration of placebo, cortisol, or yohimbine (to raise noradrenaline levels) depending on group assignment. On Day 3, all words presented on Day 1 were presented. As expected, on Day 3, memory was significantly enhanced for associations that were cued and successfully retrieved on Day 2 compared to uncued associations. However, for associative d', there was no Cued × Group interaction nor a main effect of Group, i.e., on the standard measure of memory performance, post-retrieval drug presence on Day 2 did not affect memory reconsolidation. As further evidence for a null result, fMRI univariate analyses showed no Cued × Group interactions in whole-brain or ROI activity.Strengths:There are some aspects of this study that I find impressive. The study is well-designed and the fMRI analysis methodology is innovative and sound. The authors have made meticulous and thorough physiological measurements, and assays of mood, throughout the experiment. By doing so, they have overcome, to a considerable extent, the difficulties inherent in the timing of human oral drug delivery in reconsolidation tasks, where it is difficult to have the drug present in the immediate recall period without affecting recall itself. This is beautifully shown in Figure 3. I also think that having some neurobiological assay of memory reactivation when studying reconsolidation in humans is critical, and the authors provide this. While multi-voxel patterns of hemodynamic responses are, in my view, very difficult to equate with an "engram", these patterns do have something to do with memory.

We thank the reviewer for considering aspects of our work impressive, the study to be well-designed, and the methodology to be innovative and sound.

Weaknesses:I have major issues regarding the behavioral results and the framing of the manuscript.(1) To arrive at group differences in memory performance, the authors performed median splitting of Day 3 trials by short and long reaction times during memory cueing on Day 2, as they took this as a putative measure of high/low levels of memory reactivation. Associative category hits on Day 3 showed a Group by Day 2 Reaction time (short, long) interaction, with post-hocs showing (according to the text) worse memory for short Day 2 RTs in the Yohimbine group. These post-hocs should be corrected for multiple comparisons, as the result is not what would be predicted (see point 2). My primary issue here is that we are not given RT data for each group, nor is the median splitting procedure described in the methods. Was this across all groups, or within groups? Are short RTs in the yohimbine group any different from short RTs in the other two groups? Unfortunately, we are not given Day 2 picture category memory levels or reaction times for each group. This is relevant because (as given in Supplemental Table S1) memory performance (d´) for the Yohimbine group on Day 1 immediate testing is (roughly speaking) 20% lower than the other 2 groups (independently of whether the pairs will be presented again the following day). I appreciate that this is not significant in a group x performance ANOVA but how does this relate to later memory performance? What were the group-specific RTs on Day 1? So, before the reader goes into the fMRI results, there are questions regarding the supposed drug-induced changes in behavior. Indeed, in the discussion, there is repeated mention of subsequent memory impairment produced by yohimbine but the nature of the impairment is not clear.

Thank you for the opportunity to clarify these important issues.

Reaction times are well established proxies (correlates) of memory strength and memory confidence in previous research, as they reflect cognitive processes involved in retrieving information. Faster reaction times indicate stronger mnemonic evidence and higher confidence in the accuracy of a memory decision, while slower responses suggest weaker evidence and decision uncertainty or doubt. This relationship is supported by an extensive literature (e.g., Starns 2021; Robinson et al., 1997; Ratcliff & Murdock, 1976; amongst others). Importantly, distinguishing between high and low confidence choices in a memory task serves the purpose of differentiating between particularly strong memory evidence (e.g., in associative cued recall, when remembering is particularly vivid) and weaker memory evidence. Separating low from high confidence responses based on participants’ reaction times was especially important in the current analyses, because previous research demonstrates that reaction times during cued recall tasks inversely correlate with hippocampal involvement Heinbockel et al., 2024; Gagnon et al. 2019 and that stress-effects on human memory may be particularly pronounced for high-confidence memories (Gagnon et al., 2019).

In response to the Reviewer 1’s comments, we have elaborated on our rationale for the distinction between short and long reaction times in the introduction, results, and methods. Please see page 4, lines 144 to 148:

“We distinguished between responses with short and long reaction times indicative of high and low confidence responses because previous research showed that reaction times are inversely correlated with hippocampal memory involvement(58-60) and memory strength(61,62), and that high confidence memories associated with short reaction times may be particularly sensitive to stress effects(63).”

On page 13, lines 520 to 523:

“Reaction times in the Day 2 Memory cueing task revealed a trial-specific gradient in reactivation strength. Thus, we turned to single-trial analyses, differentiating Day 3 trials by short and long reaction times during memory cueing on Day 2 (median split), indicative of high vs. low memory confidence(58–60) and hippocampal reactivation(26,63).”

And on page 26, lines 1046 to 1053:

“Reaction times serve as a proxy for memory confidence and memory strength, with faster responses reflecting higher confidence/strength and slower responses suggesting greater uncertainty/weaker memory. The association between reaction times and memory confidence has been established by previous research(58–60), suggesting that the distinction between high from low confidence responses differentiates vividly recalled associations from decisions based on weaker memory evidence. Reaction times are further linked to hippocampal activity during recall tasks(26,53), and stress effects on memory are particularly pronounced for high-confidence memories(53).”

With respect to behavioral data reporting, we agree that the critical median-split procedure was not sufficiently clear in the original manuscript. We elaborate on this important aspect of the analysis now on page 26, lines 1053 to 1057:

“We conducted a median-split within each participant to categorize trials as fast vs. slow reaction time trials during Day 2 memory cueing. We conducted this split on the participant- and not group-level because there is substantial inter-individual variability in overall reaction times. This approach also results in an equal number of trials in the low and high confidence conditions.”

We completely agree that the relevant post-hoc test should be corrected for multiple comparisons. Please note that all reported post-hoc tests had been Bonferroni-corrected already. We clarify this now by explicitly referring to corrected p-values (P_*corr*_) and indicate in the methods that P_*corr*_ refers to Bonferroni-corrected p-values. (please see page 25, lines 1036 to 1038).

We further agree that for a comprehensive overview of the behaviour in terms of memory performance and RTs, these data need to be provided for each group and experimental day. Therefore, we now extended Supplementary Table S1 to include descriptive indices of memory performance (hits, dprime) and RTs for each group for each day. Moreover, we now report ANOVAs for reaction times for each of the experimental days in the main text.

The ANOVA for Day 1 is now reported on page 6, lines 200 to 204: “To test for potential group differences in reaction times for correctly remembered associations on Day 1, we fit a linear model including the factors *Group* and *Cueing*. Critically, we did not observe a significant *Group* x *Cueing* interaction, suggesting no RT difference between groups for later cued and not cued items (F(2,58) = 1.41, P = .258, η^2^ = 0.01; Supplemental Table S1).”

The ANOVA for Day 2 is now reported on page 7, lines 243 to 248: “To test for potential group differences in reaction times for correctly remembered associations on Day 2, we fit a linear model including the factors *Group* and *Reaction time (slow/fast)* following the subject specific median split. The model did not reveal any main effect or interaction including the factor *Group* (all Ps > .535; Supplemental Table S1), indicating that there was no RT difference between groups, nor between low and high RT trials in the groups.”

The ANOVA for Day 3 is reported on page 13 lines 487 to 494: “To test for potential group differences in reaction times for correctly remembered associations on Day 3 we fit a linear model including the factors *Group* and *Cueing*. This model did not reveal any main effect or interaction including the factor *Group* (all Ps > .267), indicating that there was no average RT difference between groups. As expected we observed a main effect of the factor *Cueing*, indicating a significant difference of reaction times across groups between trials that were successfully cued and those not cued on Day 2 (F(2,58) = 153.07, P < .001, η^2^ = 0.22; Supplemental Table S1).”

(2) The authors should be clearer as to what their original hypotheses were, and why they did the experiment. Despite being a complex literature, I would have thought the hypotheses would be reconsolidation impairment by cortisol and enhancement by yohimbine. Here it is relevant to point out that - only when the reader gets to the Methods section - there is mention of a paper published by this group in 2024. In this publication, the authors used the same study design but administered a stress manipulation after Day 2 cued recall, instead of a pharmacological one. They did not find a difference in associative hit rate between stress and control groups, but - similar to the current manuscript - reported that post-retrieval stress disrupts subsequent remembering (Day 3 performance) depending on neural memory reinstatement during reactivation (specifically driven by the hippocampus and its correlation with neocortical areas).Instead of using these results, and other human studies, to motivate the current work, reference is made to a recent animal study: Line 169 "Building on recent findings in rodents (Khalaf et al. 2018), we hypothesized that the effects of post-retrieval noradrenergic and glucocorticoid activation would critically depend on the reinstatement of the neural event representation during retrieval". It is difficult to follow that a rodent study using contextual fear conditioning and examining single neuron activity to remote fear recall and extinction would be relevant enough to motivate a hypothesis for a human psychopharmacological study on emotionally neutral paired associates.

We agree that our recent publication utilizing a very similar experimental design including three days is highly relevant in the context of the current study and we now refer to this recent study earlier in our manuscript. Please see page 3, lines 89 to 94:

“Recently, we showed a detrimental impact of post-retrieval stress on subsequent memory that was contingent upon reinstatement dynamics in the Hippocampus, VTC and PCC during memory reactivation26. While this study provided initial insights into the potential brain mechanisms involved in the effects of post-retrieval stress on subsequent memory, the underlying neuroendocrine mechanisms remained elusive.”

Moreover, we explicitly state our hypothesis regarding the neural mechanism, with reference to our recent work, on page 5, lines 166 to 169:

“Building on our recent findings in humans(26) as well as current insights from rodents(47), we hypothesized that the effects of post-retrieval noradrenergic and glucocorticoid activation would critically depend on the reinstatement of the neural event representation during retrieval.”

Concerning the potential direction of the effects of post-retrieval cortisol and noradrenaline, the literature is indeed mixed with partially contradicting results, which made it, in our view, difficult to derive a clear hypothesis of potentially opposite effects of cortisol and yohimbine. We summarize the relevant evidence in the introduction on pages 3 to 4, lines 100 to 113:

“Some studies, using emotional recognition memory or fear conditioning in healthy humans, suggest enhancing effects of post-retrieval glucocorticoids on subsequent memory(30,31). However, rodent studies on neutral recognition memory(21), fear conditioning(32), as well as evidence from humans on episodic recognition memory(33) report impairing effects of glucocorticoid receptor activation on post-retrieval memory dynamics. For noradrenaline, post-retrieval blockade of noradrenergic activity impairs putative reconsolidation or future memory accessibility in human fear conditioning(34), as well as drug (alcohol) memory(35) and spatial memory in rodents(36). However, this effect is not consistently observed in human studies on fear conditioning(40), speaking anxiety(37), inhibitory avoidance(39), traumatic mental imagination (PTSD patients)(38), and might depend on the arousal state of the individual(21) or the exact timing of drug administration as suggested by studies in humans(41) and rodents(42). Thus, while there is evidence that glucocorticoid and noradrenergic activation after retrieval can affect subsequent memory, the direction of these effects remains elusive.”

In addition to these reviewer comments and in response to the eLife assessment, we would like to emphasize that the present findings are in our view not only relevant for a subfield but may be of considerable interest for researchers from various fields, beyond experimental memory research, including Neurobiology, Psychiatry, Clinical Psychology, Educational Psychology, or Law Psychology. We highlight the relevance of the topic and our findings now more explicitly in the introduction and discussion. Please see page 3:

“The dynamics of memory after retrieval, whether through reconsolidation of the original trace or interference with retrieval-related traces, have fundamental implications for educational settings, eyewitness testimony, or mental disorders(5,11,12). In clinical contexts, post-retrieval changes of memory might offer a unique opportunity to retrospectively modify or render less accessible unwanted memories, such as those associated with posttraumatic stress disorder (PTSD) or anxiety disorders(13–15). Given these potential far reaching implications, understanding the mechanisms underlying post-retrieval dynamics of memory is essential.”

On page 17:

“Upon their retrieval, memories can become sensitive to modification(1,2). Such post-retrieval changes in memory may be fundamental for adaptation to volatile environments and have critical implications for eyewitness testimony, clinical or educational contexts(5,11–15). Yet, the brain mechanisms involved in the dynamics of memory after retrieval are largely unknown, especially in humans.”

And on page 19:

“Beyond their theoretical relevance, these findings may have relevant implications for attempts to employ post-retrieval manipulations to modify unwanted memories in anxiety disorders or PTSD(97,98). Specifically, the present findings suggest that such interventions may be particularly promising if combined with cognitive or brain stimulation techniques ensuring a sufficient memory reactivation.“

**Reviewer #1 (Recommendations for the authors):**
(1) Related to major issue 2 in the Public Review. In the introduction, it would be helpful to be specific about the type of memory being probed in the different studies referenced (episodic vs conditioning). For the former, please make it clear whether stimuli to be remembered were emotional or neutral, and for which stimulus class drug effects were observed. This is particularly important given that in the first paragraph, you describe memory reactivation in the context of traumatic memories via mention of PTSD. It would also be helpful to know to which species you refer. For example, in line 115, "timing of drug administration..." a rodent and a human study are cited.

We completely agree that these aspects are important. We have therefore rewritten the corresponding paragraph in the introduction to clarify the type of memory probed, the emotionality of the stimuli and the species tested. Please see pages 3 to 4, lines 100 to 113:

“Some studies, using emotional recognition memory or fear conditioning in healthy humans, suggest enhancing effects of post-retrieval glucocorticoids on subsequent memory(30,31). However, rodent studies on neutral recognition memory(21), fear conditioning(32), as well as evidence from humans on episodic recognition memory(33) report impairing effects of glucocorticoid receptor activation on post-retrieval memory dynamics. For noradrenaline, post-retrieval blockade of noradrenergic activity impairs putative reconsolidation or future memory accessibility in human fear conditioning(34), as well as drug (alcohol) memory(35) and spatial memory in rodents(36). However, this effect is not consistently observed in human studies on fear conditioning(40), speaking anxiety(37), inhibitory avoidance(39), traumatic mental imagination (PTSD patients)(38), and might depend on the arousal state of the individual(21) or the exact timing of drug administration as suggested by studies in humans(41) and rodents(42). Thus, while there is evidence that glucocorticoid and noradrenergic activation after retrieval can affect subsequent memory, the direction of these effects remains elusive.”

(2) The Bos 2014 reference appears incorrect. I think you mean the Frontiers paper of the same year.

Thank you for noticing this mistake, which has been corrected.

(3) Line 734 "The study employed a fully crossed, placebo-controlled, double-blind, between-subjects design". What is a fully crossed design?

A fully-crossed design refers to studies in which all possible combinations of multiple between-subjects factors are implemented. However, because the factor reactivation/cueing was manipulated within-subject in the present study and there is only one between-subjects factor (group/drug), “fully-crossed” may be misleading here. We removed it from the manuscript.

(4) Supplemental Table S3. Are these ordered in terms of significance? A t- or Z-value for each cluster (either of the peak or a summed value) would be helpful.

We agree that the ordering of the clusters was not clearly described. In the revised Supplemental Table S3, we have now added a column with the cluster-peak specific T-values and added an explanation in the table caption: “Depicted clusters are ordered by cluster-peak T-values.”

(5) Please provide the requested memory performance and reaction time data, and relevant group comparisons.

In response to general comment #1 above, we now provide all relevant accuracy and reaction time data for all groups and experimental days in the revised Supplemental Table S1. Moreover, we now report the relevant group comparisons in the main text on page 6, lines 200 to 204, on page 7, lines 243 to 248, and on page 13, lines 487 to 494.

(6) Please rewrite the introduction with specific hypotheses, mention your recent results published in Science Advances, and attend to suggestions made in the first comment above.

We have rewritten parts of the introduction to make the link to our recent publication clearer and to clarify the types of memories and species tested, as suggested by the reviewer (please see pages 3 to 4, lines 100 to 113). Moreover, we explicitly state our hypothesis regarding the neural mechanism on page 5, lines 166 to 169:

“Building on our recent findings in humans(26) as well as current insights from rodents(47), we hypothesized that the effects of post-retrieval noradrenergic and glucocorticoid activation would critically depend on the reinstatement of the neural event representation during retrieval.”

In terms of the direction of the potential cortisol and yohimbine effects, we have elaborated on the relevant literature, which in our view does not allow a clear prediction regarding the nature of the drug effects. We have made this explicit by stating that “… while there is evidence that glucocorticoid and noradrenergic activation after retrieval can affect subsequent memory, the direction of these effects remains elusive.” (please see page 4, lines 111 to 113). It would be, in our view, inappropriate to retrospectively add another, more specific “hypothesis”.

**Reviewer #2 (Public review):**
Summary:The authors aimed to investigate how noradrenergic and glucocorticoid activity after retrieval influence subsequent memory recall with a 24-hour interval, by using a controlled three-day fMRI study involving pharmacological manipulation. They found that noradrenergic activity after retrieval selectively impairs subsequent memory recall, depending on hippocampal and cortical reactivation during retrieval.Overall, there are several significant strengths of this well-written manuscript.Strengths:(1) The study is methodologically rigorous, employing a well-structured three-day experimental design that includes fMRI imaging, pharmacological interventions, and controlled memory tests.(2) The use of pharmacological agents (i.e., hydrocortisone and yohimbine) to manipulate glucocorticoid and noradrenergic activity is a significant strength.(3) The clear distinction between online and offline neural reactivation using MVPA and RSA approaches provides valuable insights into how memory dynamics are influenced by noradrenergic and glucocorticoid activity distinctly.

We thank the reviewer for these very positive and encouraging remarks.

Weaknesses:(1) One potential limitation is the reliance on distinct pharmacodynamics of hydrocortisone and yohimbine, which may complicate the interpretation of the results.

We agree that the pharmacodynamics of hydrocortisone and yohimbine are different. However, we took these pharmacodynamics into account when designing the experiment and have made an effort to accurately track the indicators for noradrenergic arousal and glucocorticoids across the experiment. As shown in Figure 2, these indicators confirm that both drugs are active within the time window of approximately 40-90 minutes after reactivation. This time window corresponds to the proposed reconsolidation window, which is assumed to open around 10 minutes post-reactivation and to remain open for a few hours (approximately 90 minutes; Monfils & Holmes, 2018; Lee et al., 2017; Monfils et al., 2009).

We have now acknowledged the distinct pharmacodynamics of hydrocortisone and yohimbine on page 21, lines 845 to 847: “We note that yohimbine and hydrocortisone follow distinct pharmacodynamics(104,105), yet selected the administration timing to ensure that both substances are active within the relevant post-retrieval time window.”

In the results section, on page 11, lines 437 to 439, we further emphasize this differential dynamic: “Our data demonstrate that, despite the distinct pharmacodynamics of CORT and YOH, both substances are active within the time window that is critical for potential reconsolidation effects(3,4,43).”

(2) Another point related above, individual differences in pharmacological responses, physiological and cortisol measures may contribute to memory recall on Day 3.

The administered drugs elicit a pronounced adrenergic and glucocorticoid response, respectively. Specifically, the cortisol levels reached by 20mg of hydrocortisone correspond to those observed after a significant stressor exposure. Moreover, individual variation in stress system activation following drug intake tends to be less pronounced than in response to a natural stressor. Nevertheless, we fully agree that individual factors, such as metabolism or body weight, can influence the drug's action.

We therefore re-analysed the reported Day 3 models, now including individual measures of baseline-to-peak changes in cortisol and systolic blood pressure, respectively. We report these additional analyses in the supplement and refer the interested reader to these analyses on page 15, lines 580 to 586:

“As individual factors, such as metabolism or body weight, can influence the drug's action, we ran an additional analysis in which we included individual (baseline-to-peak) differences in salivary cortisol and (systolic) blood pressure, respectively. This analysis did not show any group by baseline-to-peak difference interaction suggesting that the observed memory effects were mainly driven by the pharmacological intervention group per se and less by individual variation in responses to the drug (see Supplemental Results).”

And in the Supplemental Results:

“To account for individual differences in cortisol responses after pill intake, we fit additional GLMMs predicting Day 3 subsequent memory of cued and correct trials including the factors *Individual baseline-to-peak cortisol* and *Group*. Doing so allowed us to account for variation in Day 3 performance, which might have resulted from within-group variation in cortisol responses, in particular in the CORT group. Importantly, none of the models predicting Day 3 memory performance by Day 2 cortisol-increase and Group, median-split RTs (high/low), hippocampal activity and RTs, or hippocampal activity and VTC category reinstatement revealed a significant group x baseline-to-peak cortisol interaction (all Ps > .122). These results suggest that inter-individual differences in cortisol responses did not have a significant impact on subsequent memory, beyond the influence of group per se. The same analyses were repeated for systolic blood pressure employing GLMMs predicting Day 3 subsequent memory of cued and correct trials including the factors *Individual baseline-to-peak systolic blood pressure* and *Group* to account for variation in Day 3 performance, which might have resulted from within-group variation in blood pressure response, in particular in the YOH group. While the model predicting Day 3 memory performance revealed a significant *Individual baseline-to-peak systolic blood pressure* × *Group* × *median-split RTs (high/low)* interaction (β = -0.05 ± 0.02, z = -2.04, P = .041, R^2^_conditional_ = 0.01), post-hoc slope tests, however, did not show any significant difference between groups (all P_Corr_ > .329). The remaining models including hippocampal activity and RTs, or hippocampal activity and VTC category reinstatement did not reveal a significant *Group* × *Individual baseline-to-peak systolic blood pressure* interaction (all Ps > .101). These results suggest that inter-individual differences in systolic blood pressure responses did not have a significant impact on subsequent memory, beyond the influence of group per se.”

Although we acknowledge that our study may not have been sufficiently powered for an analysis of individual differences, these data suggest that our memory effects were mainly driven by the pharmacological intervention group per se and less by individual variation in responses. It is to be noted, however, that all participants of the respective groups showed a pronounced increase in cortisol concentrations (on average > 1000% in the CORT group) and autonomic arousal (on average > 10% in the YOH group), respectively. These increases appeared to be sufficient to drive the observed memory effects, irrespective of some individual variation in the magnitude of the response.

(3) Median-splitting approach for reaction times and hippocampal activity should better be justified.

Reaction times are well established proxies (correlates) of memory strength and memory confidence in previous research, as they reflect cognitive processes involved in retrieving information. Faster reaction times indicate stronger mnemonic evidence and higher confidence in the accuracy of a memory decision, while slower responses suggest weaker evidence and decision uncertainty or doubt. This relationship is supported by an extensive literature (e.g., Starns 2021; Robinson et al., 1997; Ratcliff & Murdock, 1976; amongst others). Importantly, distinguishing between high and low confidence choices in a memory task serves the purpose to differentiating between particularly strong memory evidence (e.g., is associative cued recall, when remembering is particularly vivid) and weaker memory evidence. Separating low from high confidence responses based on participants’ reaction times was especially important in the current analyses, because previous research demonstrates that reaction times during cued recall tasks inversely correlate with hippocampal involvement Heinbockel et al., 2024; Gagnon et al. 2019 and that stress-effects on human memory may be particularly pronounced for high-confidence memories (Gagnon et al., 2019).

In response to the Reviewer comments, we have elaborated on our rationale for the distinction between short and long reaction times in the introduction, results, and methods. Please see page 4, lines 144 to 148:

“We distinguished between responses with short and long reaction times indicative of high and low confidence responses because previous research showed that reaction times are inversely correlated with hippocampal memory involvement(58–60) and memory strength(61,62), and that high confidence memories associated with short reaction times may be particularly sensitive to stress effects(63).”

On page 13, lines 520 to 523:

“Reaction times in the Day 2 Memory cueing task revealed a trial-specific gradient in reactivation strength. Thus, we turned to single-trial analyses, differentiating Day 3 trials by short and long reaction times during memory cueing on Day 2 (median split), indicative of high vs. low memory confidence(58–60) and hippocampal reactivation(26,63).”

And on page 26, lines 1046 to 1053:

“Reaction times serve as a proxy for memory confidence and memory strength, with faster responses reflecting higher confidence/strength and slower responses suggesting greater uncertainty/weaker memory. The association between reaction times and memory confidence has been established by previous research(58–60), suggesting that the distinction between high from low confidence responses differentiates vividly recalled associations from decisions based on weaker memory evidence. Reaction times are further linked to hippocampal activity during recall tasks(26,53), and stress effects on memory are particularly pronounced for high-confidence memories(53).”

We agree that the critical median-split procedure was not sufficiently clear in the original manuscript. We elaborate on this important aspect of the analysis now on page 26, lines 1053 to 1057:

“We conducted a median-split within each participant to categorize trials as slow vs. fast reaction time trials during Day 2 memory cueing. We chose to conduct this split on the participant- and not group-level because there is substantial inter-individual variability in overall reaction times and to retain an equal number of trials in the low and high confidence conditions.”

In addition to these reviewer comments and in response to the eLife assessment, we would like to emphasize that the present findings are in our view not only relevant for a subfield but may be of considerable interest for researchers from various fields, beyond experimental memory research, including Neurobiology, Psychiatry, Clinical Psychology, Educational Psychology, or Law Psychology. We highlight the relevance of the topic and our findings now more explicitly in the introduction and discussion. Please see page 3:

“The dynamics of memory after retrieval, whether through reconsolidation of the original trace or interference with retrieval-related traces, have fundamental implications for educational settings, eyewitness testimony, or mental disorders5,11,12. In clinical contexts, post-retrieval changes of memory might offer a unique opportunity to retrospectively modify or render less accessible unwanted memories, such as those associated with posttraumatic stress disorder (PTSD) or anxiety disorders(13–15). Given these potential far reaching implications, understanding the mechanisms underlying post-retrieval dynamics of memory is essential.”

On page 17:

“Upon their retrieval, memories can become sensitive to modification(1,2). Such post-retrieval changes in memory may be fundamental for adaptation to volatile environments and have critical implications for eyewitness testimony, clinical or educational contexts(5,11–15), Yet, the brain mechanisms involved in the dynamics of memory after retrieval are largely unknown, especially in humans.”

And on page 19:

“Beyond their theoretical relevance, these findings may have relevant implications for attempts to employ post-retrieval manipulations to modify unwanted memories in anxiety disorders or PTSD(97,98). Specifically, the present findings suggest that such interventions may be particularly promising if combined with cognitive or brain stimulation techniques ensuring a sufficient memory reactivation.“

**Reviewer #2 (Recommendations for the authors):**
My comments and/or questions for the authors to improve this well-written manuscript.(1) This study identifies the modulatory role of the hippocampus and VTC in the effects of norepinephrine on subsequent memory. Are there functional interactions between these ROIs and other brain regions that could be wise to consider for a more comprehensive understanding of the underlying neural mechanisms?

We agree that functional interactions of hippocampus and VTC and other regions that were active during Day 2 memory cueing are relevant for our understanding of the underlying mechanisms. We therefore now performed connectivity analyses using general psycho-physiological interaction analysis (gPPI; as implemented in SPM) and report the results of this analysis on page 16, lines 635 to 644, and added Supplemental Table S4 including gPPI statistics.

“We conducted general psycho-physiological interaction analysis (gPPI) analyses on the Day 2 memory cueing task (remembered – forgotten), which revealed that successful cueing was accompanied by significant functional connectivity between the left hippocampus, VTC, PCC and MPFC (see Supplemental Table S4). However, using these connectivity estimates to predict Day 3 subsequent memory performance (dprime) via regression did not reveal any significant *Group* × *Connectivity* interactions, indicating that the pharmacological manipulation (i.e. noradrenergic stimulation) did not modulate subsequent memory based on functional connectivity during memory cueing (all P_Corr_ > .228). The same pattern of results was observed when including single trial beta estimates from multiple ROIs during memory cueing to predict Day 3 memory (all interaction effects P_Corr_ > .288).”

(2) In theory, noradrenergic activity would have a profound impact on activity in widespread brain regions that are closely related to memory function. It would be interesting to know other possible effects beyond the hippocampus and VTC.

We agree and included in our analysis additional ROIs beyond the HC and VTC; we now report these explorative results on page 16, lines 616 to 633:

“Beyond hippocampal and VTC activity during memory cueing (Day 2), we exploratively reanalysed the GLMMs predicting Day 3 memory performance including the PCC, which was relevant during memory cueing in the current study and in our previous work(26). Predicting Day 3 memory performance by the factors *Group* and *Single trial beta activity* during memory cueing in the PCC did not reveal a significant interaction (P_Corr_ = 1); adding the factor *Reaction time* to the model also did not result in a significant interaction (P_Corr_ = 1). We also included the Medial Prefrontal Cortex (MPFC) to predict Day 3 memory performance, as the MPFC has been shown to be sensitive to noradrenergic modulation in previous work(75). Predicting Day 3 memory performance by the factors *Group* and *Single trial beta activity* during memory cueing in the MPFC did not reveal a significant interaction (P_Corr_ = 1); adding the factor *Reaction time* to the model also did not result in a significant interaction (P_Corr_ = 1), which indicates that the MPFC was not modulated by either pharmacological intervention. Finally, we investigated memory cueing from all remaining ROIs that were significantly activated during the Day 2 memory cueing task (Day 2 whole-brain analysis; correct-incorrect; Supplemental Table S3). We again fit GLMMs predicting Day 3 memory performance by the factors *Group* and *Single trial beta activity* during memory cueing. Again, we did not observe any significant interaction effect any of the ROIs (all interaction P_Corr_ > .060) and these results did not change when adding the factor *Reaction time* to the respective models (all P_Corr_ > .075).”

(3) There are substantial individual differences in pharmacological responses, physiological and cortisol measures, as shown in Figure 3A&B. If such individual differences are taken into account, are there any potential effects on subsequent recall on Day 3 pertaining to the hydrocortisone group?

In response to this comment (and the General comment #1 of this reviewer), we now re-analyzed the respective models including individual measures of baseline-to-peak cortisol and systolic blood pressure.

We re-analysed the reported Day 3 models, now including individual measures of baseline-to-peak changes in cortisol and systolic blood pressure, respectively. We report these additional analyses in the supplement and refer the interested reader to these analyses on page 15, lines 580 to 586:

“As individual factors, such as metabolism or body weight, can influence the drug's action, we ran an additional analysis in which we included individual (baseline-to-peak) differences in salivary cortisol and (systolic) blood pressure, respectively. This analysis did not show any group by baseline-to-peak difference interaction suggesting that the observed memory effects were mainly driven by the pharmacological intervention group per se and less by individual variation in responses to the drug (see Supplemental Results).”

And in the Supplemental Results:

“To account for individual differences in cortisol responses after pill intake, we fit additional GLMMs predicting Day 3 subsequent memory of cued and correct trials including the factors *Individual baseline-to-peak cortisol* and *Group*. Doing so allowed us to account for variation in Day 3 performance, which might have resulted from within-group variation in cortisol responses, in particular in the CORT group. Importantly, none of the models predicting Day 3 memory performance by Day 2 cortisol-increase and Group, median-split RTs (high/low), hippocampal activity and RTs, or hippocampal activity and VTC category reinstatement revealed a significant group x baseline-to-peak cortisol interaction (all Ps > .122). These results suggest that inter-individual differences in cortisol responses did not have a significant impact on subsequent memory, beyond the influence of group per se. The same analyses were repeated for systolic blood pressure employing GLMMs predicting Day 3 subsequent memory of cued and correct trials including the factors *Individual baseline-to-peak systolic blood pressure* and *Group* to account for variation in Day 3 performance, which might have resulted from within-group variation in blood pressure response, in particular in the YOH group. While the model predicting Day 3 memory performance revealed a significant *Individual baseline-to-peak systolic blood pressure* × *Group* × *median-split RTs (high/low)* interaction (β = -0.05 ± 0.02, z = -2.04, P = .041, R^2^_conditional_ = 0.01), post-hoc slope tests, however, did not show any significant difference between groups (all P_Corr_ > .329). The remaining models including hippocampal activity and RTs, or hippocampal activity and VTC category reinstatement did not reveal a significant *Group* × *Individual baseline-to-peak systolic blood pressure* interaction (all Ps > .101). These results suggest that inter-individual differences in systolic blood pressure responses did not have a significant impact on subsequent memory, beyond the influence of group per se.”

(4) Median-splitting approach for reaction times and hippocampal activity should better be justified.

Reaction times are well established proxies (correlates) of memory strength and memory confidence in previous research, as they reflect cognitive processes involved in retrieving information. Faster reaction times indicate stronger mnemonic evidence and higher confidence in the accuracy of a memory decision, while slower responses suggest weaker evidence and decision uncertainty or doubt. This relationship is supported by an extensive literature (e.g., Starns 2021; Robinson et al., 1997; Ratcliff & Murdock, 1976; amongst others). Importantly, distinguishing between high and low confidence choices in a memory task serves the purpose to differentiating between particularly strong memory evidence (e.g., is associative cued recall, when remembering is particularly vivid) and weaker memory evidence. Separating low from high confidence responses based on participants’ reaction times was especially important in the current analyses, because previous research demonstrates that reaction times during cued recall tasks inversely correlate with hippocampal involvement Heinbockel et al., 2024; Gagnon et al. 2019 and that stress-effects on human memory may be particularly pronounced for high-confidence memories (Gagnon et al., 2019).

In response to the Reviewer comments, we have elaborated on our rationale for the distinction between short and long reaction times in the introduction, results, and methods. Please see page 4, lines 144 to 148:

“We distinguished between responses with short and long reaction times indicative of high and low confidence responses because previous research showed that reaction times are inversely correlated with hippocampal memory involvement(58–60) and memory strength(61,62), and that high confidence memories associated with short reaction times may be particularly sensitive to stress effects(63).”

On page 13, lines 520 to 523:

“Reaction times in the Day 2 Memory cueing task revealed a trial-specific gradient in reactivation strength. Thus, we turned to single-trial analyses, differentiating Day 3 trials by short and long reaction times during memory cueing on Day 2 (median split), indicative of high vs. low memory confidence(58–60) and hippocampal reactivation(26,63).”

And on page 26, lines 1046 to 1053:

“Reaction times serve as a proxy for memory confidence and memory strength, with faster responses reflecting higher confidence/strength and slower responses suggesting greater uncertainty/weaker memory. The association between reaction times and memory confidence has been established by previous research(58–60), suggesting that the distinction between high from low confidence responses differentiates vividly recalled associations from decisions based on weaker memory evidence. Reaction times are further linked to hippocampal activity during recall tasks(26,53), and stress effects on memory are particularly pronounced for high-confidence memories(53).”

Minor comments:(5) Please include the full names of key abbreviations in the figure legends, such as "ass.cat.hit" and among others.

We now include the full names of key abbreviations in all figure legends (e.g., ass.cat.hit = associative category hit).

(6) Please introduce various metrics used in the study to aid readers in better understanding the measurements they utilized.

We agree that various measures that were included in our analyses had not been described clearly enough before, especially concerning the multivariate analyses. We therefore added short explanations across the results section.

Page 8, lines 279 to 280: “Classifier accuracy is derived from the sum of correct predictions the trained classifier made in the test-set, relative to the total amount of predictions.”

Page 8, lines 290 to 292: “Neural reinstatement reflects the extent to which a neural activity pattern (i.e., for objects) that was present during encoding is reactivated during retrieval (e.g., memory cueing).”

Page 8, lines 299 to 301: “The logits here reflect the log-transformed trial-wise probability of a pattern either representing a scene or an object.”

Page 10, lines 378 to 380: “Beyond category-level reinstatement, we assessed event-level memory trace reinstatement from initial encoding (Day 1) to memory cueing (Day 2), via RSA, correlating neural patterns in each region (hippocampus, VTC, and PCC) across days.”

(7) Please explain what the different colors represent in Figures 5B and 5C to avoid confusion. It would be good to indicate significant differences in the figures if applicable.

We now added line legends to the figure and also the caption to clarify what exactly is depicted. We added asterisks to mark significant differences.

References:

Monfils, M. H., Cowansage, K. K., Klann, E., & LeDoux, J. E. (2009). Extinction-reconsolidation boundaries: key to persistent attenuation of fear memories. *science*, *324*(5929), 951-955.

Monfils, M. H., & Holmes, E. A. (2018). Memory boundaries: opening a window inspired by reconsolidation to treat anxiety, trauma-related, and addiction disorders. *The Lancet Psychiatry*, *5*(12), 1032-1042.

Lee, J. L. C., Nader, K. & Schiller, D. An Update on Memory Reconsolidation Updating. *Trends Cogn. Sci.* 21, 531–545 (2017).

Radley, J. J., Williams, B., & Sawchenko, P. E. (2008). Noradrenergic innervation of the dorsal medial prefrontal cortex modulates hypothalamo-pituitary-adrenal responses to acute emotional stress. *Journal of Neuroscience*, *28*(22), 5806-5816.

Heinbockel, H., Wagner, A. D., & Schwabe, L. (2024). Post-retrieval stress impairs subsequent memory depending on hippocampal memory trace reinstatement during reactivation. *Science Advances*, *10*(18), eadm7504.